# Gradient Descent with Large Step Size Restores Symmetry in Deep Linear Networks with Multi-Pathway

Hee-Sung Kim [1]   Sungyoon Lee [1]

## Abstract

Recent analyses of multi-pathway Deep Linear Networks use Gradient Flow to predict a "winner-takes-all" specialization in which path symmetry breaks and each feature concentrates in a single pathway. In this work, we show that discrete Gradient Descent (GD) with a large step size tells a different story. We prove that single-path solutions are sharp minima, whereas distributing signals across pathways reduces sharpness by a factor that decreases with both the number of pathways and depth. Consequently, while early training reproduces the depth-driven symmetry breaking predicted by GF, oscillations at the Edge of Stability subsequently override this tendency and drive the network into a re-balancing phase, where signals redistribute across pathways. Together, these results clarify how depth shapes pathway competition and explain why large-step GD favors shared representations rather than persistent single-pathway dominance.

## 1. Introduction

Understanding the training dynamics of Deep Neural Networks remains a central challenge in deep learning theory. Although deep architectures are highly expressive, how optimization algorithms select specific solutions among many global minima—the *implicit bias* of the optimizer—remains poorly understood (Zhang et al., 2017; Neyshabur et al., 2017; Gunasekar et al., 2017; Soudry et al., 2018). To characterize this bias, a growing body of work analyzes Deep Linear Networks (DLNs) (Saxe et al., 2014; Arora et al., 2018; Lampinen & Ganguli, 2019; Shi et al., 2022), under the Gradient Flow (GF) approximation, which assumes an infinitesimally small learning rate.

Recently, Shi et al. (2022) utilized the GF to argue that multi-pathway DLNs undergo "symmetry breaking," converging to a sparse "winner-takes-all" solution where parallel over-parameterization becomes redundant. However, this continuous-time analysis overlooks the discrete dynamics of Gradient Descent (GD) with large learning rates, which characterize practical training at the Edge of Stability (Cohen et al., 2021). Given that GD actively interacts with the loss curvature to avoid sharp minima, it remains a critical question whether the symmetry-breaking phenomenon persists under GD with large learning rates.

In this work, we demonstrate that the answer is no: while GF predicts symmetry breaking, the discreteness of GD induces a fundamentally different phenomenon we term **pathway re-balancing**. By analyzing the geometry of the loss landscape, we prove that sparse, single-path solutions correspond to sharp minima, whereas solutions balanced across multiple pathways are significantly flatter. Consequently, when the learning rate is sufficiently large, the implicit regularization of GD drives the network away from unstable, sparse configurations toward stable, balanced configurations.

Our main contributions are summarized as follows:

- **Sharpness Reduction via Parallelism:** We theoretically derive the relationship between the number of pathways $H$, depth $L$, and sharpness of global minima. We prove that balancing signals across $H$ pathways reduces sharpness by $H^{2/L-1}$ (Theorem 4.2).

- **Re-balancing Dynamics at the Edge of Stability:** We identify two distinct phases of training under large step sizes. Initially, the network exhibits symmetry breaking similar to GF. However, as the dominant path sharpens, the model enters a **re-balancing** phase. Here, violent oscillations force the network to flatten its landscape by distributing signals across multiple paths.

- **Worst-Case Return Threshold**: Based on a *deep linear chain* model, we derive a depth-dependent upper bound on learning rate that guarantees the trajectory survives violent oscillations. The ratio of this threshold to the classical stability limit increases with depth, broadening the window for re-balancing.

[1]Department of Computer Science, Hanyang University, Seoul, Korea. Correspondence to: Sungyoon Lee <sungyoon-lee@hanyang.ac.kr>.

*Proceedings of the 43rd International Conference on Machine Learning*, Seoul, South Korea. PMLR 306, 2026. Copyright 2026 by the author(s).

Beyond the multi-pathway setting, our results suggest that architectural biases predicted under Gradient Flow—a foundation shared by many recent analyses of DLN-based models—may require re-examination once discrete dynamics and finite learning rates are accounted for. This highlights the essential role of discrete optimization in shaping the final network structure and motivates revisiting GF-based predictions through the lens of GD.

## 2. Related Work

**Deep Linear Networks and Multi-Pathway Dynamics.** Deep linear networks (DLNs) serve as a canonical testbed for isolating optimization and representation learning mechanisms (Baldi & Hornik, 1989; Saxe et al., 2014; Arora et al., 2018). They have been extensively used to study exact training dynamics and emergent phenomena (Saxe et al., 2019; Lampinen & Ganguli, 2019; J Dominé et al., 2023; Nam et al., 2025). There has been recent attention to multi-branch architectures. Notably, Shi et al. (2022) and Saxe et al. (2022) demonstrate that under continuous-time Gradient Flow (GF), parallel pathways exhibit a "winner-takes-all" dynamic. Specifically, they identify that some pathways dominate the feature learning driven either by small initialization asymmetries (Shi et al., 2022) or by structural asymmetry such as depth or shared representations across pathways (Saxe et al., 2022).

**Implicit Bias of GD and Edge of Stability.** In contrast to GF, discrete-time Gradient Descent (GD) with large learning rates induces distinct dynamical regimes, often operating at the "Edge of Stability" (EoS) (Cohen et al., 2021; Wu et al., 2018). Theoretical and empirical studies suggest that instability in this regime acts as an implicit regularizer, driving models toward flatter minima (Damian et al., 2023) or inducing transient "catapult" phases that favor generalization (Lewkowycz et al., 2020; Zhu et al., 2024). Recent works have formalized these discrete dynamics through mechanisms like self-stabilization (Damian et al., 2023) and central flows (Cohen et al., 2025), highlighting how GD avoids the sharpest directions in the landscape.

**GD in Linear Models.** Recent research has begun to bridge these insights specifically within linear models. Marion & Chizat (2024) and Even et al. (2023) show that GD in linear networks explicitly favors flat minima, differing from the implicit bias of GF. Most relevant to our work, Ghosh et al. (2025) analyze deep matrix factorization beyond the stability threshold, identifying oscillatory behaviors and convergence properties that violate GF predictions. Our work extends these findings to the multi-pathway setting of Shi et al. (2022), demonstrating that the stability constraints of large-step GD override the symmetry-breaking tendency of GF, forcing a *re-balancing* of features across pathways to reduce sharpness.

## 3. Problem Setup

We study a deep linear network with $H$ parallel pathways indexed by $h \in [H] = \{1, \cdots, H\}$. Pathway $h$ has depth $L_h$ and weight matrices $W_{h\ell} \in \mathbb{R}^{d \times d}$ for layers $\ell \in [L_h]$. The architecture follows the multi-pathway formulation of Shi et al. (2022). The end-to-end map of pathway $h$ is the ordered product

$$\Omega_h := W_{hL_h} W_{hL_h - 1} \cdots W_{h1}. \tag{1}$$

The overall input–output map of the network is the sum of the pathway maps:

$$M := \sum_{h=1}^{H} \Omega_h. \tag{2}$$

As in prior theoretical work on deep matrix factorization (Ghosh et al., 2025), we focus on the square case $W_{h\ell} \in \mathbb{R}^{d \times d}$ for theoretical analysis.

### 3.1. Optimization Objective

Let $\Theta = \{W_{h\ell}\}_{h \in [H], \ell \in [L_h]}$ denote all parameters. We fit a target matrix $M_\star \in \mathbb{R}^{d \times d}$ by minimizing the squared Frobenius loss

$$\mathcal{L}(\Theta) = \frac{1}{2} \|M - M_\star\|_F^2. \tag{3}$$

We consider discrete-time gradient descent updates

$$W_{h\ell}(t+1) = W_{h\ell}(t) - \eta \nabla_{W_{h\ell}} \mathcal{L}(\Theta(t)), \tag{4}$$

with stepsize $\eta > 0$.

### 3.2. Target-Aligned Parametrization and the SVS Set

Let the target admit the SVD

$$M_\star = U_\star \Sigma_\star V_\star^\top, \qquad \Sigma_\star = \text{diag}(\sigma_{\star 1}, \cdots, \sigma_{\star d}). \tag{5}$$

A standard simplification in the analysis of deep linear networks aligns each pathway's end-to-end singular vectors with those of the target, so the dynamics decouple across modes (Saxe et al., 2014). We adopt the multi-pathway extension of this parametrization, the *singular vector stationary* (SVS) set of Ghosh et al. (2025). The SVS set consists of parameter configurations $\Theta$ for which each layer factors as

$$W_{h\ell} = Q_{h\ell+1} \Sigma_{h\ell} Q_{h\ell}^\top, \; Q_{h1} = V_\star, \; Q_{hL_h+1} = U_\star, \tag{6}$$

with fixed orthogonal matrices $\{Q_{h,\ell}\}$ and diagonal $\Sigma_{h\ell}$. Under (6), adjacent orthogonals cancel in the product, so every pathway map is diagonal in the target basis:

$$\Omega_h = U_\star \Big( \prod_{\ell=1}^{L_h} \Sigma_{h\ell} \Big) V_\star^\top. \tag{7}$$

Gradient descent preserves the SVS set: if $\Theta(t)$ lies in the set, so does $\Theta(t+1)$ (Appendix E.1). Analogous target-aligned reductions appear in Saxe et al. (2014; 2022); Arora et al. (2019); Gidel et al. (2019); Varre et al. (2023); Chou et al. (2024); Kwon et al. (2024).

We index modes by the target singular vector pairs $(u_{\star i}, v_{\star i})$, the $i$-th columns of $U_\star$ and $V_\star$, and track the scalar *mode coefficients*

$$\sigma_{hi}(t) := u_{\star i}^\top \Omega_h(t) \, v_{\star i}, \tag{8}$$

$$\sigma_i(t) := u_{\star i}^\top M(t) \, v_{\star i} = \sum_{h=1}^{H} \sigma_{hi}(t). \tag{9}$$

On the SVS set, $\sigma_{hi}(t)$ equals the $i$-th singular value of $\Omega_h(t)$, and the learning dynamics decouple across modes $i \in [d]$.

Mode-wise decoupling on the SVS set decomposes the loss into independent scalar mode losses:

$$\mathcal{L}(\Theta) = \sum_{i=1}^{d} \mathcal{L}_i, \quad \mathcal{L}_i = \frac{1}{2}\Big(\sum_{h=1}^{H} \sigma_{hi} - \sigma_{\star i}\Big)^2. \tag{10}$$

### 3.3. Depth-Balanced Initialization

We initialize each pathway with

$$W_{h\ell}(0) = \alpha_h^{1/L_h} \, I_d, \tag{11}$$

where $\alpha_h > 0$ are small pathway-dependent scales. This initialization places $\Theta(0)$ in the SVS set and equalizes the layerwise singular values within each pathway. It therefore lies on the *depth-balanced manifold*, defined by

$$\sigma_{hi}^{(\ell)} = \sigma_{hi}^{1/L_h}, \qquad \forall\, h \in [H],\, \ell \in [L_h],\, i \in [d], \tag{12}$$

where $\sigma_{hi}^{(\ell)}$ denotes the $i$-th singular value of layer $\ell$ in pathway $h$. The trajectory remains on this manifold (and in the SVS set) throughout training for symmetric targets $M_\star = V_\star \Sigma_\star V_\star^\top$ (Proposition B.4 in Appendix B.2). The scales $\alpha_h$ may differ across pathways. Such tiny initialization asymmetries trigger the symmetry-breaking dynamics analyzed in Section 5.

# 4. Sharpness Reduction in Multi-Pathway Deep Linear Models

In this section, we derive the relationship between the number of pathways $H$ and the loss sharpness at global minima in depth-balanced manifold. We assume homogeneous depth $L_h = L$ in this section. We demonstrate that distributing a target feature across multiple pathways significantly reduces sharpness. Detailed proofs for all results are provided in the Appendix C.

### 4.1. Sharpness Analysis

The sharpness of the loss is determined by the maximum eigenvalue of the Hessian $\nabla^2 \mathcal{L}(\Theta)$ at a global minimum where $M = M_\star$. On the SVS set (Section 3.2), the Hessian decomposes across modes, so we can analyze the Hessian contribution from each mode $i \in [d]$ independently.

**Proposition 4.1** (Mode-Wise Dominant Eigenvalue). *At any global minimum within the SVS set restricted to the depth-balanced manifold, the Hessian contribution from mode $i$ has a single nonzero eigenvalue given by:*

$$\lambda_i = L \sum_{h=1}^{H} \sigma_{hi}^{2-\frac{2}{L}}. \tag{13}$$

Since the Hessian on this manifold is block-diagonal across modes with rank-one blocks, the loss sharpness equals $\lambda_{\max} = \max_{i \in [d]} \lambda_i$.

This result highlights that sharpness depends on how the total singular value $\sigma_{\star i}$ is distributed among the $H$ pathways. We now state the main theorem regarding the optimal distribution of these values.

**Theorem 4.2** (Sharpness Reduction via Parallelism). *Assume $L > 2$. Among all global minima satisfying the constraint $\sum_{h=1}^{H} \sigma_{hi} = \sigma_{\star i}$, the mode-wise sharpness $\lambda_i$ is minimized when the target singular value is distributed equally across all $H$ pathways:*

$$\sigma_{1i} = \cdots = \sigma_{Hi} = \frac{\sigma_{\star i}}{H}. \tag{14}$$

*The minimal eigenvalue on the depth-balanced manifold is given by:*

$$\lambda_i^{\min} = L \, H^{\frac{2}{L}-1} \, \sigma_{\star i}^{2-\frac{2}{L}}. \tag{15}$$

Theorem 4.2 implies a significant reduction in sharpness compared to a single-pathway model. Comparing the balanced multi-pathway sharpness to that of a single pathway ($\lambda_i^{\min}(H=1)$), we obtain the reduction factor:

$$\frac{\lambda_i^{\min}}{\lambda_i^{\min}(H=1)} = H^{\frac{2}{L}-1} < 1. \tag{16}$$

Since $2/L - 1 < 0$ for $L > 2$, increasing either $H$ or $L$ strictly decreases this ratio.

Under the conventional ordering $\sigma_{\star 1} \geq \sigma_{\star 2} \geq \cdots \geq \sigma_{\star d}$, the largest target mode dominates the sharpness: $\max_i \lambda_i^{\min} = \lambda_1^{\min}$. The reduction factor $H^{2/L-1}$ applies every mode, so re-balancing benefits any mode $i$ whose single-path sharpness $S_i = L\sigma_{\star i}^{2-2/L}$ exceeds the stability threshold $2/\eta$ (Section 5).

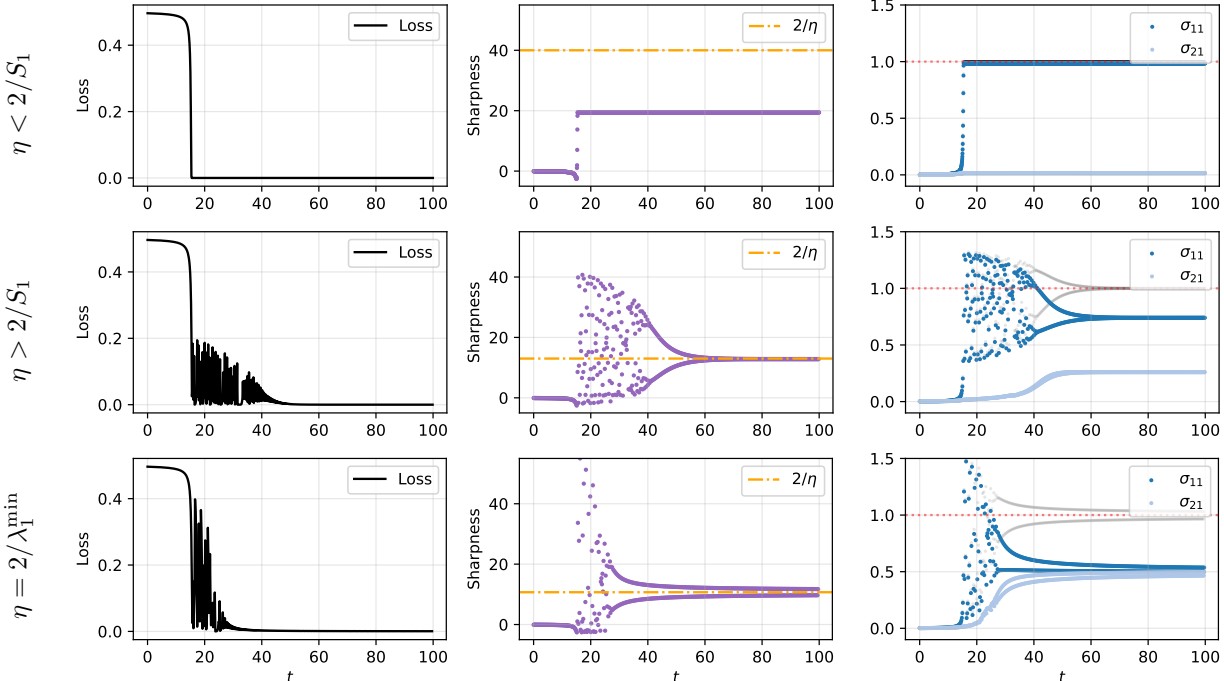

*Figure 1.* Training dynamics of GD with multi-path DLNs ($L = 20$, $H = 2$, $\sigma_{\star 1} = 1$) across different learning rates $\eta$. The evolution of pathway singular values is shown in blue ($\sigma_{11}$, $h = 1$), light blue ($\sigma_{21}$, $h = 2$), and gray ($\sigma_1 = \sigma_{11} + \sigma_{21}$). While small $\eta$ leads to a single-path solution, GD with a larger learning rate ($\eta > 2/S_1$) drives the system toward a more balanced configuration where singular values are distributed across pathways. Consequently, the sharpness $\lambda_{\max}$ (purple dots) is suppressed and converges to $2/\eta$.

## 5. Dynamics of GD Beyond Classical Stability

We now study how multi-pathway DLNs learn when $\eta$ exceeds the stability threshold of the sharpest minima. The dynamics reflect two opposing forces: the architectural bias of GF toward "winner-takes-all" specialization, and the implicit bias of large-step GD toward flat minima. Small learning rates let the former dominate and reproduce GF's symmetry breaking; large learning rates let the latter override it, driving the network into a distinct *re-balancing* phase.

### 5.1. Two Stages of Training with Large Learning Rate

We focus on mode $i = 1$ with homogeneous depth $L$, where $S_i = L\sigma_{\star i}^{2-2/L}$ is the sharpness of the "winner-takes-all" solution carrying the entire target in a single pathway ($\sigma_{11} = \sigma_{\star 1}$ and $\sigma_{h1} = 0$ for $h \neq 1$). To trigger symmetry breaking, we initialize each pathway with a slightly different scale ($\alpha_1 > \alpha_2 > \cdots > \alpha_H$).

**Small Learning Rate Regime** ($\eta < 2/S_1$). When the learning rate is small, the discrete GD dynamics closely approximate the continuous GF. In this regime, small differences in initialization lead to the "winner-takes-all" symmetry breaking (Shi et al., 2022): the relative growth rate $\dot\sigma_{hi}/\dot\sigma_{ki} = (\sigma_{hi}/\sigma_{ki})^{L-1}$ amplifies infinitesimal initial asymmetries through the depth-$L$ exponent, so the pathway

with the largest initialization for a given mode grows exponentially faster than the others and suppresses competing gradients. As shown in the first row of Figure 1, the network converges to a dominant-path solution ($\sigma_{11} = \sigma_{\star 1}$) while other pathways stay near zero, yielding a sharp minimum ($\lambda_{\max} \approx S_1$).

**Large Learning Rate Regime** ($\eta > 2/S_1$). Training with a learning rate large enough to destabilize the single-path solution exhibits two distinct phases:

- **Phase 1: Symmetry Breaking.** Early in training, the singular values are small, so local sharpness stays below $2/\eta$. The dynamics track GF, and the same depth-$L$ amplification mechanism that drives the small-LR regime concentrates the signal in the dominant pathway.

- **Phase 2: Re-balancing (Drift via Oscillations).** As the $\sigma_{11}$ approaches $\sigma_{\star 1}$, the sharpness rises near $S_1$ and eventually exceeds $2/\eta$. GD can no longer settle into this sharp minimum and enters the Edge of Stability regime, where residuals oscillate around the loss surface rather than decay. These oscillations are not noise: they drive a systematic transfer of signal from the dominant pathway to the others, reducing sharpness toward the stability threshold.

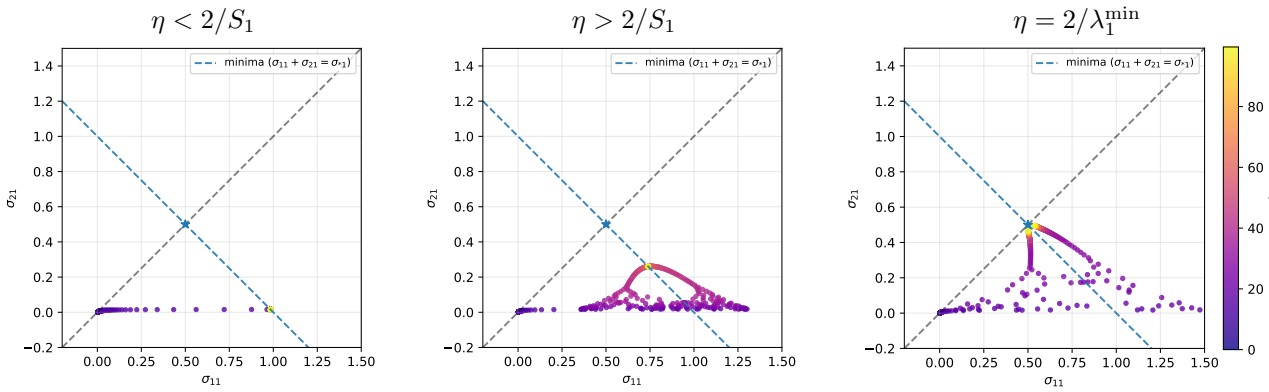

*Figure 2.* Trajectories of $(\sigma_{11}, \sigma_{21})$ for different learning rates $\eta$. In the stable regime (left), the dominant pathway suppresses others. Beyond the stability threshold (middle, right), the pathways bifurcate into the more balanced minimum ($\lambda_{\max} < S_1$).

Appendix E identifies the local mechanism driving this phase. On the SVS depth-balanced manifold, the dynamics reduce to a scalar recursion per mode and pathway. Under gradient flow, a conserved quantity fixes the leaf coordinate $z$ that measures pathway imbalance, and the sharpness at any zero-loss point grows quadratically in $\|z\|$. Large-step GD at the Edge of Stability breaks the conservation law: under local self-stabilization, the residual alternates in sign, and $\|z\|$ decays over pairs of steps. The oscillations thus act as a directed drift toward flatter, more balanced minima.

This drift acts only while the sharpness exceeds $2/\eta$, so the trajectory re-balances until the $\sigma_{hi}$ are distributed enough to enter the stability window

$$2/S_1 < \eta < 2/\lambda_1^{\min}.$$

By Theorem 4.2, this balanced configuration is a significantly flatter minimum, $\lambda_1^{\min} = H^{2/L-1}S_1$. At $\eta = 2/\lambda_1^{\min}$ the stopping boundary coincides with the fully balanced minimum; at intermediate rates the trajectory halts at a partially balanced configuration. Instability at the Edge of Stability therefore acts not as noise but as a directed force that selects flatter, pathway-balanced minima (Figures 2).

**Nonlinear Setting.** While our theory assumes linearity, we observe the same transition in multi-pathway MLPs with Tanh activation (Figure 3). Following Shi et al. (2022), we use the canonical basis for inputs $x$ and initialize the weights from $\mathcal{N}(0, \frac{0.02}{n_{\text{in}}+n_{\text{out}}})$. We monitor learning dynamics via the projection $K_{h1} = u_1^\top \Omega_h v_1$. In the stable regime ($\eta \approx 1/S_1$), we recover the "winner-takes-all" symmetry breaking of linear networks. A large learning rate ($\eta \approx 2/\lambda_1^{\min}$) instead drives the system toward a flatter, balanced minimum ($K_{11} \approx K_{21}$). suggesting that the re-balancing mechanism extends to nonlinear architectures. Figure 7 in Appendix G shows results at additional learning rates; MLPs typically need slightly higher rates than linear networks to fully balance.

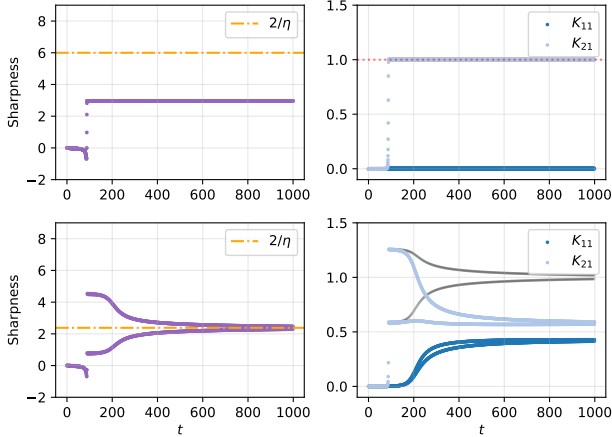

*Figure 3.* **Symmetry Breaking vs. Re-balancing in Nonlinear Networks.** Training dynamics of a two-pathway MLP ($L = 3$, $H = 2$, Tanh activation). Training with small $\eta$ (Top row) shows symmetry breaking and Training with larger $\eta = 2/\lambda_1^{\min}$ (Bottom row) shows re-balancing phase.

### 5.2. Contrast with Depth-Wise Balancing

A fundamental distinction exists between the pathway re-balancing observed in this work and the depth-wise balancing analyzed in single-pathway networks. Ghosh et al. (2025) demonstrate that large learning rates minimize the layer-wise balancing gap, which is contingent on explicit initialization asymmetries and serves only to rectify unbalanced initial conditions. In standard Deep Linear Networks (DLNs), layers naturally remain aligned throughout Gradient Flow (GF) trajectories due to conservation laws (e.g., $W_{\ell+1}W_{\ell+1}^\top - W_\ell^\top W_\ell = \text{const}$). The depth-wise balancing is only prominent with highly unbalanced initializations $W_{\ell+1}W_{\ell+1}^\top - W_\ell^\top W_\ell \gg 0$. However, for balanced or zero initializations ($W_{\ell+1}W_{\ell+1}^\top - W_\ell^\top W_\ell \approx 0$), GF and GD trajectories remain qualitatively indistinguishable regarding their balancing dynamics.

In contrast, symmetry breaking in multi-pathway architectures is a spontaneous phenomenon. As shown by Shi et al. (2022), GF dynamics inherently amplify infinitesimal asymmetries, driving the system toward a dominant-path solution ("winner-takes-all"). Unlike depth-wise imbalance, which depends on specific initial conditions, pathway imbalance is the natural attractor of continuous-time dynamics. Our results demonstrate that discrete-time instability actively counteracts this intrinsic bias. Specifically, it destabilizes the sharp, single-path minima favored by GF, thereby forcing the network into a balanced configuration that continuous dynamics cannot achieve.

### 5.3. Heterogeneous Depth between Pathways

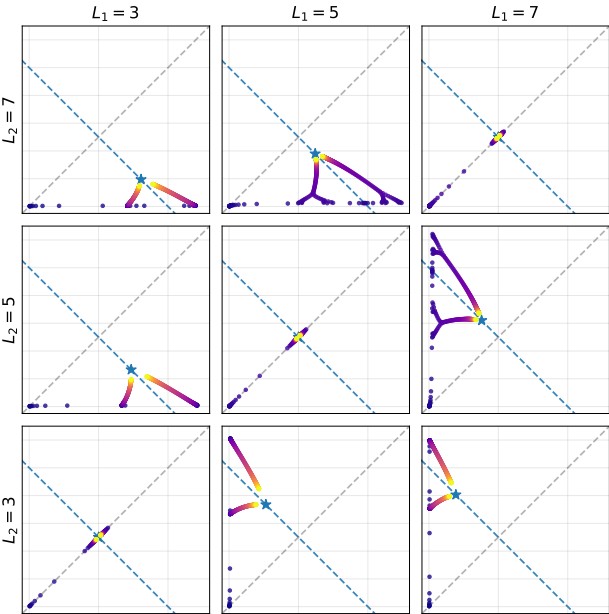

*Figure 4.* Trajectories of $(\sigma_{11}, \sigma_{21})$ for heterogeneous depth models with learning rate $\eta^* = 2/\lambda_1^{\min}$. See Figure 2 together.

So far we assumed homogeneous depth $L$, where only initialization differences trigger symmetry breaking. We now allow heterogeneous depths $L_h$ across pathways. Different depths break symmetry structurally: even under balanced initialization, gradient magnitudes scale with depth, so pathways accelerate at different rates.

We first extend the sharpness–allocation relationship to heterogeneous depths.

**Proposition 5.1** (Sharpness with Heterogeneous Depth).
*Consider a multi-pathway network where pathway $h$ has depth $L_h$. On the depth-balanced manifold, the dominant eigenvalue of the Hessian for mode $i$ is given by:*

$$\lambda_i = \sum_{h=1}^{H} L_h\, \sigma_{hi}^{2 - \frac{2}{L_h}}. \qquad (17)$$

Unlike the homogeneous case, equal splitting ($\sigma_{hi} = \sigma_{\star i}/H$) no longer minimizes sharpness. The Lagrange conditions for minimizing $\lambda_i$ subject to $\sum_h \sigma_{hi} = \sigma_{\star i}$ give:

$$2(L_h - 1)\sigma_{hi}^{1 - \frac{2}{L_h}} = \mu, \quad \forall h \in [H], \qquad (18)$$

where $\mu$ is the Lagrange multiplier enforcing the constraint. The condition equalizes $\partial \lambda_i / \partial \sigma_{hi}$ across pathways. The shallow pathway still carries the largest share of $\sigma_{\star i}$ but the optimum spreads more mass into deeper pathways than the "winner-takes-all" solution. These equations lack a closed form, but we can solve them numerically for any $\{L_h\}$ to obtain the optimal allocation $\sigma_{hi}^{\mathrm{opt}}$ and minimum sharpness $\lambda_i^{\min}$.

**Kinetic bias vs. stability bias.** Heterogeneous depth provides a stringent test of our theory. Under GF (or small $\eta$), the shallower pathway learns faster and absorbs the signal (Saxe et al., 2022), so structural asymmetry alone drives "winner-takes-all" toward the shallowest path. But Proposition 5.1 shows that concentrating the full signal in a shallow pathway yields a sharper minimum than concentrating it in a deep one, since $2 - 2/L_h$ shrinks as $L_h$ decreases. GF thus favors shallow paths kinetically, while large-step GD favors distributed paths for stability—the two forces conflict.

Training with large $\eta$ resolves this conflict in two phases:

- **Early Phase (Structural Symmetry Breaking):** Near initialization, the per-pathway growth rate scales as $\sigma_{hi}^{(L_h-1)/L_h}$, which is larger for smaller $L_h$ when $\sigma_{hi} \ll 1$. The shorter pathway (e.g., $L_1 = 3$) receives larger gradients and grows exponentially faster than the deeper pathway (e.g., $L_2 = 7$), driving the system toward a shallow-path-dominated solution.

- **Late Phase (Re-balancing):** Once the shallow path captures most of the signal, local sharpness climbs past $2/\eta$ and the path destabilizes. The induced oscillations (Section 5.1) keep the residual nonzero, which drives the deeper, flatter pathway upward until the configuration reaches the stability window.

Figure 4 shows the dynamics for $(L_1, L_2) \in \{3, 5, 7\}^2$ at $\eta = 2/\lambda_1^{\min}$ (from our numerical solution). We initialize all pathways with the same scale to isolate architectural effects. The trajectories converge to the numerically predicted equilibrium that equalizes marginal sharpness across pathways. Large-step GD therefore overrides structural asymmetry and restores a balanced representation, even when architecture favors a single pathway. See Figure 9 in Appendix G for results across a wider depth range $(L_1, L_2) \in \{3, 4, 5, 6, 7\}^2$.

# 6. A Worst-Case Return Threshold from a Deep Linear Chain (Role of Depth)

Re-balancing demands a large step size: to destabilize the sharp single-path minimum, $\eta$ must exceed $2/S_1$, where $S_1 = L\sigma_{\star 1}^{2-2/L}$ is the dominant mode's sharpness; pushing $\eta$ higher unseats more modes, balancing the top $p$ (made precise below). But $\eta$ has a ceiling. Re-balancing first passes through a transient single-path phase, and an overlarge step there drives the dominant pathway's singular value across zero—a sign flip that breaks the SVS description and aborts re-balancing. The failure is worst when the signal concentrates in a single pathway: a *deep linear chain*, from which we derive a depth-dependent ceiling $\eta_{\mathrm{WCR}}$. The resulting overshoot-but-return window, measured relative to the onset scale, widens with depth, so deeper networks tolerate the large steps required for re-balancing.

**The binding constraint: the transient single pathway.** During symmetry breaking (Section 5), the dominant mode concentrates in one pathway while the others remain near zero. This dominant pathway is the sharpest component of the landscape and therefore sets the tightest stability constraint. Hence, a step size large enough to induce re-balancing can drive it into violent oscillations. To keep the SVS description valid, the dominant coordinate must remain positive: crossing $u_{\star 1}^\top \Omega_h v_{\star 1} \leq 0$ corresponds to a singular-vector sign flip across the origin saddle. Thus, the binding question is whether the dominant coordinate survives its overshoot.

**Reduction to a deep linear chain.** On the SVS depth-balanced manifold, per-layer scalars $a_{h1} \geq 0$ carry mode 1, with end-to-end value $\sigma_{h1} = a_{h1}^L$ (Appendix F). When the mode concentrates in a single pathway, the contributions of the other pathways become negligible in the residual, and the dynamics collapse to one scalar—the *deep-chain map*.

$$w_{t+1} = f_\eta(w_t) := w_t - \eta\, w_t^{L-1}\big(w_t^L - \sigma_{\star 1}\big), \quad (19)$$

where $w_t \equiv a_{11}(t)$ with target fixed point $w_\star = \sigma_{\star 1}^{1/L}$. The $w^{L-1}$ prefactor makes the update highly sensitive to the current per-layer scale: once the iterate overshoots $w_\star$, the next correction can be large enough to cross the origin.

**The worst-case return threshold.** We ask that the chain recover from *any* overshoot without crossing the origin.

**Definition 6.1** (Worst-case return threshold). With $w_\star = \sigma_{\star 1}^{1/L}$ and $f_\eta$ from (19), a stepsize $\eta$ is *two-step return-safe* if every overshoot from $(0, w_\star)$ returns to the interval $(0, w_\star)$ after one further step:

$$\forall\, w \in (0, w_\star): \ f_\eta(w) > w_\star \implies 0 < f_\eta^{\circ 2}(w) < w_\star.$$

The *worst-case return threshold* $\eta_{\mathrm{WCR}}(L, \sigma_{\star 1})$ is the supremum of all return-safe stepsizes.

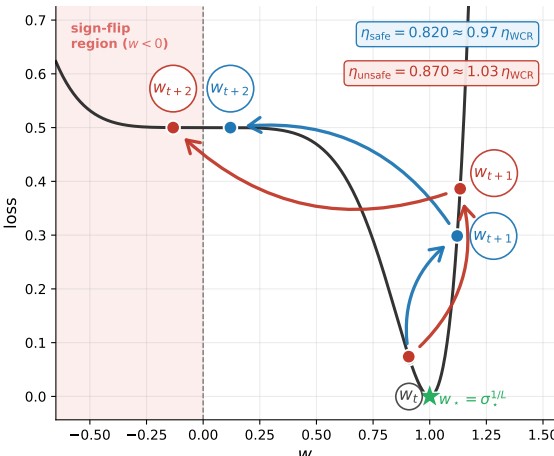

*Figure 5.* Loss $\frac{1}{2}\big(w^L - \sigma_{\star 1}\big)^2$ of the deep linear chain ($L = 5$, $\sigma_{\star 1} = 1$; $w$ is the per-layer scale, minimizer $w_\star = \sigma_{\star 1}^{1/L} = 1$). GD follows the map (19). From $w_t$, the maximizer of $f_\eta$ producing the largest overshoot $w_{t+1}$, the second step $w_{t+2}$ either returns to $(0, w_\star)$ (blue, $\eta \approx 0.97\,\eta_{\mathrm{WCR}}$) or crosses the origin into $w < 0$ (red, $\eta \approx 1.03\,\eta_{\mathrm{WCR}}$). $\eta_{\mathrm{WCR}}$ is the knife-edge between the two.

On the relevant overshoot branch, the binding case is the largest overshoot, produced by the maximizer of $f_\eta$ (Appendix F); if that worst overshoot returns, every smaller one does too. Figure 5 shows the knife-edge: at $\eta \approx 0.97\,\eta_{\mathrm{WCR}}$ the iterate bounces back into $(0, w_\star)$, while at $\eta \approx 1.03\,\eta_{\mathrm{WCR}}$ it overshoots the origin into the sign-flip region.

**Closed-form scaling and the role of depth.** The threshold separates into a curvature scale and a depth factor. Writing $\gamma := \eta S_1$—so the overshoot-onset scale $\eta_1 = 1/S_1$ is $\gamma = 1$ and the classical stability bound $\eta_{\mathrm{stable}} = 2/S_1$ is $\gamma = 2$—we have

$$\eta_{\mathrm{WCR}}(L, \sigma_{\star 1}) = \frac{\gamma_{\mathrm{WCR}}(L)}{S_1}, \quad (20)$$

with $\gamma_{\mathrm{WCR}}(L)$ depending only on depth.

**Theorem 6.2** (Existence and computation). *For every integer $L \geq 2$ there is a unique $\gamma_{\mathrm{WCR}}(L) > 2$ for which (20) satisfies Definition 6.1. It is the root of a single scalar equation, computable by bisection (Appendix F).*

Since $\gamma_{\mathrm{WCR}}(L) > 2$, the safe ceiling always exceeds the classical bound $\eta_{\mathrm{stable}} = 2/S_1$: the chain tolerates steps beyond the classical linear-stability threshold. The margin over the onset scale is exactly this constant,

$$\frac{\eta_{\mathrm{WCR}}}{\eta_1} = \gamma_{\mathrm{WCR}}(L), \quad (21)$$

independent of the target $\sigma_{\star 1}$.

**Proposition 6.3** (Growth with depth). *The normalized WCR ratio $\gamma_{\mathrm{WCR}}(L) = \eta_{\mathrm{WCR}}/\eta_1$ grows unboundedly with depth and admits the asymptotic scaling*

$$\gamma_{\mathrm{WCR}}(L) = \Theta(\log L). \tag{22}$$

*In particular, deeper chains admit a strictly larger window of "overshoot-but-return" learning rates:*

$$\eta_1 < \eta < \eta_{\mathrm{WCR}}(L, \sigma_{\star 1})$$
$$\text{with} \quad \frac{\eta_{\mathrm{WCR}}}{\eta_1} = \gamma_{\mathrm{WCR}}(L) \uparrow \text{ as } L \uparrow, \tag{23}$$

*so that the maximal safe overshoot amplitude (in the worst-case sense of Definition 6.1) increases with depth.*

This is also visible in the return-map view in Appendix F (Figure 6): at a fixed ratio $\eta/\eta_1$, the worst overshoot's next iterate stays positive as $L$ grows, turning a sign flip at small depth into a safe return at large depth.

**The admissible window and rank-$p$ balancing.** We can now identify which modes re-balance. On the depth-balanced manifold, mode $i$ has single-path sharpness $S_i$ and flattest balanced sharpness $\lambda_i^{\min} = H^{2/L-1} S_i$ (Theorem 4.2), and leaves the single-path attractor once $\eta > 2/S_i$. Since $S_1 \geq \cdots \geq S_d$, a step in the band

$$\frac{2}{S_p} < \eta < \min\left\{\frac{2}{S_{p+1}}, \eta_{\max}\right\},$$
$$\eta_{\max} := \min\left\{\frac{2}{\lambda_1^{\min}}, \eta_{\mathrm{WCR}}\right\}, \tag{24}$$

re-balances the top $p$ modes and leaves the tail in the GF-like regime; the achievable rank is the largest $p$ with $2/S_p < \eta_{\max}$. The two ceilings in $\eta_{\max}$ guard different phases: $2/\lambda_1^{\min}$ keeps the *balanced* attractor stable, while $\eta_{\mathrm{WCR}}$ keeps the dominant path alive through the *transient*. In normalized units they read $\gamma = 2H^{1-2/L}$ and $\gamma = \gamma_{\mathrm{WCR}}(L)$; both increase with depth, and which one binds depends on $H$ and $L$. Either way, depth raises $\eta_{\max}$, and with it the number of modes that re-balance.

**A mechanism for warm-up.** The ceiling also hints at why learning-rate warm-up helps. Sharpness peaks early, while the dominant path is still forming, so a step large enough to re-balance later may exceed $\eta_{\mathrm{WCR}}$ during that peak and trigger a sign flip. warm-up resolves the tension: a small early step carries the trajectory safely through the sharp transient, and a later increase sustains the drift toward flat, multi-pathway minima. In this view, warm-up does not change the re-balancing mechanism; it delays the large-step regime until the trajectory has passed the most fragile single-path bottleneck.

## 7. Discussion

Large-step Gradient Descent qualitatively changes the multi-pathway learning predicted by continuous-time theories. Beyond our specific setting, this finding affects how we interpret a broader class of GF-based architectural bias results.

Theoretical accounts of over-parameterized models often predict that training selects sparse, isolated substructures. GF analyses of multi-pathway DLNs predict "winner-takes-all" specialization (Shi et al., 2022; Saxe et al., 2022), and the Lottery Ticket Hypothesis (Frankle & Carbin, 2019) conjectures that trained over-parameterized networks contain sparse subnetworks carrying most of their function. Our results suggest that, even within such a subnetwork view, large-step GD's preference for low-curvature minima can reshape which substructures emerge: rather than concentrating signal in a single pathway, GD tends to spread it across parallel ones. How this flat-minima bias interacts with the formation of sparse subnetworks under large-step GD is an open question worth exploring.

GF-based analyses also argue that structural asymmetries (e.g., depth differences) drive the system toward a winner-takes-all solution favoring the shallower path (Saxe et al., 2022). Section 5.3 shows that large-step GD overrides such asymmetries within the range of depths we test: once the sharp single-path minimum becomes GD-unstable, the system moves toward a flatter, distributed configuration even when GF would predict a strongly asymmetric solution. In this regime, the optimizer's implicit bias toward flat minima outweighs the structural bias toward shallow pathways predicted by GF.

## 8. Conclusion

We studied how large-step Gradient Descent shapes learning in multi-pathway Deep Linear Networks, and showed that its behavior departs qualitatively from the continuous-time picture. First, distributing a target feature equally across $H$ parallel pathways reduces sharpness by $H^{2/L-1}$ relative to a single-path solution (Theorem 4.2): sparse minima are the sharpest configurations on the depth-balanced manifold, while balanced ones are the flattest. Second, training at the Edge of Stability proceeds in two phases. Symmetry breaking concentrates the signal in a dominant pathway, as GF predicts. Once local sharpness exceeds $2/\eta$, oscillating residuals transfer mass to subordinate pathways until the configuration satisfies the stability constraint—overriding both initialization-induced and depth-induced asymmetries. Third, a Worst-Case Return threshold derived from a deep linear chain bounds $\eta$ from above to guarantee the trajectory survives the transient oscillations. The ratio $\eta_{\mathrm{WCR}}/\eta_{\mathrm{stable}}$ grows as $\Theta(\log L)$, so deeper architectures admit a wider window of re-balancing learning rates.

Together, these results show that the symmetry breaking predicted by Gradient Flow does not persist once the infinitesimal-learning-rate assumption is relaxed. Architectural biases derived under continuous-time dynamics deserve re-examination through the lens of discrete optimization. Extending this analysis to nonlinear and modular architectures, such as Mixture-of-Experts and multi-head attention, is a natural next step.

## Impact Statement

This paper presents work whose goal is to advance the field of Machine Learning. There are many potential societal consequences of our work, none which we feel must be specifically highlighted here.

## Acknowledgments

We thank the anonymous reviewers for insightful reviews. This work was partially supported by Institute of Information & communications Technology Planning & Evaluation (IITP) grants (RS-2020-II201373, Artificial Intelligence Graduate School Program (Hanyang University); RS-2023-002206284, Artificial intelligence for prediction of structure-based protein interaction reflecting physicochemical principles); the BK21 FOUR (Fostering Outstanding Universities for Research) project; NRF2024S1A5C3A02043653, Socio-Technological Solutions for Bridging the AI Divide: A Blockchain and Federated Learning-Based AI Training Data Platform) and Korea Institute for Advanced Study (KIAS) grant funded by the Korean government (MSIT).

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

# A. Notation

The following table summarizes the key mathematical notations used throughout this paper.

| Notation | Description |
|---|---|
| $H$ | Number of parallel pathways in the network. |
| $h$ | Index for a specific pathway, $h \in [H] = \{1, \ldots, H\}$. |
| $L_h, L$ | Depth of pathway $h$, and homogeneous depth when $L_h = L$ for all $h$. |
| $d$ | Dimension of the square weight matrices ($d \times d$). |
| $W_{h\ell}$ | Weight matrix of layer $\ell \in [L_h]$ in pathway $h$. |
| $\Omega_h$ | End-to-end transformation matrix of pathway $h$ ($\Omega_h = W_{hL_h} \cdots W_{h1}$). |
| $M, M_\star$ | Overall map of the network ($M = \sum_{h=1}^{H} \Omega_h$) and the target matrix. |
| $\Theta, \Theta_h$ | Set of all network parameters, and parameters specific to pathway $h$. |
| $u_{\star i}, v_{\star i}$ | $i$-th left and right singular vectors of the target matrix $M_\star$. |
| $\sigma_{\star i}$ | $i$-th target singular value. |
| $\sigma_{hi}(t)$ | Singular mode coefficient of pathway $h$ for mode $i$ at time $t$. |
| $\sigma_i(t)$ | Total singular mode coefficient of the network for mode $i$ ($\sum_{h=1}^{H} \sigma_{hi} = u_{\star i}^\top M v_{\star i}$). |
| $\alpha_h$ | Pathway-dependent initialization scale. |
| $\mathcal{L}(\Theta), \mathcal{L}_i$ | Total squared Frobenius loss, and independent scalar mode loss for mode $i$. |
| $\eta$ | Learning rate (step size) of discrete gradient descent. |
| $\lambda_i$ | Dominant eigenvalue of the Hessian for mode $i$ (sharpness). |
| $\lambda_i^{\min}$ | Minimal eigenvalue of the Hessian on the depth-balanced manifold. |
| $S_i$ | Sharpness of a single-path ("winner-takes-all") solution for mode $i$. |
| $a_{hi}(t)$ | Per-layer scalar parameter on the depth-balanced SVS manifold ($a_{hi}^L = \sigma_{hi}$). |
| $w_t, w_\star$ | State variable representing $a_{11}(t)$ and its target fixed point ($\sigma_{\star 1}^{1/L}$) in the deep-chain map. |
| $\eta_{\text{stable}}$ | Classical linear stability bound for the learning rate ($2/S_1$). |
| $\eta_{\text{WCR}}$ | Worst-Case Return (WCR) threshold for the learning rate. |
| $\gamma, \gamma_{\text{WCR}}(L)$ | Normalized step size ($\eta S_1$) and the depth-dependent constant for the WCR threshold. |

# B. Invariance of Depthwise Balancing under the SVS Assumption

This section justifies the restriction to the depth-balanced manifold used in our sharpness analysis. Our argument follows the singular-vector-stationary (SVS) viewpoint developed for deep matrix factorization in (Ghosh et al., 2025) and adapts it to the multi-pathway architecture.

**Setting.** We consider the multi-pathway deep linear network in Section 3: there are $H$ parallel pathways. Pathway $h \in \{1, \ldots, H\}$ has depth $L_h$, layer weights $\{W_{h\ell} \in \mathbb{R}^{d \times d}\}_{\ell=1}^{L_h}$, end-to-end map $\Omega_h := W_{hL_h} \cdots W_{h1}$, total map $M := \sum_{h=1}^{H} \Omega_h$, and squared Frobenius loss

$$\mathcal{L}(\Theta) := \frac{1}{2} \|M - M_\star\|_F^2, \qquad M_\star = U_\star \Sigma_\star V_\star^\top, \ \ \Sigma_\star = \text{diag}(\sigma_{1\star}, \cdots, \sigma_{d\star}). \tag{25}$$

Throughout, we work in the (mode) coordinates induced by the fixed singular vectors $(U_\star, V_\star)$ of $M_\star$.

## B.1. SVS parametrization and mode-wise decoupling

**Definition B.1** (SVS parametrization for each pathway). We say $\Theta$ lies in the SVS set associated with $M_\star$ if for each pathway $h$ there exist *fixed* orthogonal matrices $\{Q_{h\ell}\}_{\ell=1}^{L_h+1}$ such that $Q_{h1} = V_\star$, $Q_{hL_h+1} = U_\star$, and each layer admits the decomposition

$$W_{h\ell} = Q_{h\ell+1} \Sigma_{h\ell} Q_{h\ell}^\top, \qquad \Sigma_{h\ell} = \text{diag}(\sigma_{h1}^{(\ell)}, \ldots, \sigma_{hd}^{(\ell)}), \tag{26}$$

for some diagonal $\Sigma_{h\ell}$ (with nonnegative diagonal entries, interpreted as layer singular values in the SVS chart).

Under Definition B.1, adjacent orthogonals cancel in products, so for each pathway

$$\Omega_h = W_{hL_h} \cdots W_{h1} = U_\star \Big( \prod_{\ell=1}^{L_h} \Sigma_{h\ell} \Big) V_\star^\top =: U_\star \Sigma_h V_\star^\top, \tag{27}$$

where $\Sigma_h$ is diagonal and its $i$-th diagonal entry is the end-to-end singular value in mode $i$:

$$\sigma_{hi} := \prod_{\ell=1}^{L_h} \sigma_{hi}^{(\ell)}. \tag{28}$$

Consequently,

$$M = \sum_{h=1}^{H} \Omega_h = U_\star \Big( \sum_{h=1}^{H} \Sigma_h \Big) V^{\star\top}, \tag{29}$$

so $M$ shares the singular vectors $(U_\star, V_\star)$ with $M_\star$ and the loss decouples across modes.

**Lemma B.2** (Mode-wise decoupling in the SVS set). *If $\Theta$ lies in the SVS set, then the loss decomposes as*

$$\mathcal{L}(\Theta) = \sum_{i=1}^{d} \mathcal{L}_i(\Theta), \qquad \mathcal{L}_i(\Theta) = \frac{1}{2} \Big( \sum_{h=1}^{H} \sigma_{hi} - \sigma_{\star i} \Big)^2, \tag{30}$$

*where $\sigma_{hi}$ is defined in* (28).

*Proof.* Since $M - M_\star = U_\star \big( \sum_h \Sigma_h - \Sigma_\star \big) V^{\star\top}$ and Frobenius norm is unitarily invariant,

$$\|M - M_\star\|_F^2 = \Big\| \sum_{h=1}^{H} \Sigma_h - \Sigma_\star \Big\|_F^2 = \sum_{i=1}^{d} \Big( \sum_{h=1}^{H} \sigma_{hi} - \sigma_{\star i} \Big)^2,$$

which yields the claim. $\square$

## B.2. Depthwise-balanced manifold and its invariance

**Definition B.3** (Depthwise-balanced manifold). For each pathway $h$ and mode $i$, define the end-to-end singular value $\sigma_{hi}$ as in (28). The depthwise-balanced manifold $\mathcal{B}$ is the subset of the SVS set where, for all $h$ and $i$,

$$\sigma_{hi}^{(\ell)} = \sigma_{hi}^{1/L_h} \quad \forall \ell \in \{1, \ldots, L_h\}, \qquad \text{equivalently} \qquad \sigma_{h1,i} = \cdots = \sigma_{hL_h,i}. \tag{31}$$

**Proposition B.4** (Depthwise balancing is invariant under GD in the SVS set). *Assume the iterates remain in the SVS set and are updated by gradient descent:*

$$W_{h\ell}(t+1) = W_{h\ell}(t) - \eta \, \nabla_{W_{h\ell}} \mathcal{L}(\Theta(t)), \qquad \forall h, \ell.$$

*If $\Theta(t) \in \mathcal{B}$ (Definition B.3) for some $t$, then $\Theta(t+1) \in \mathcal{B}$. Consequently, if the initialization is depthwise-balanced, then $\Theta(t) \in \mathcal{B}$ for all $t \geq 0$.*

*Moreover, letting the mode residual be*

$$r_i(t) := \sum_{h=1}^{H} \sigma_{hi}(t) - \sigma_{\star i}, \tag{32}$$

*the GD update of each layer singular value (in the SVS coordinates) is, for all $h, \ell, i$,*

$$\sigma_{hi}^{(\ell)}(t+1) = \sigma_{hi}^{(\ell)}(t) - \eta \, r_i(t) \prod_{j \in \{1,\ldots,L_h\}, \, j \neq \ell} \sigma_{h,j,i}(t). \tag{33}$$

*Proof.* Fix a pathway $h$ and layer $\ell$. The standard gradient formula for deep linear networks gives

$$\nabla_{W_{h\ell}} \mathcal{L} = W_{hL_h:\ell+1}^{\top} (M - M_\star) \, W_{h\ell-1:1}^{\top}, \tag{34}$$

where $W_{h,a:b} := W_{h,a} \cdots W_{h,b}$ with the convention $W_{h,a:b} = I$ when $a < b$.

Under Definition B.1, partial products collapse to the fixed bases with diagonal cores:

$$W_{h,L_h:\ell+1}^{\top} = Q_{h,\ell+1}\Big( \prod_{j=\ell+1}^{L_h} \Sigma_{h,j} \Big) U^{\star\top}, \qquad W_{h,\ell-1:1}^{\top} = V_\star \Big( \prod_{j=1}^{\ell-1} \Sigma_{h,j} \Big) Q_{h,\ell}^{\top}.$$

Also, since $M = U_\star(\sum_k \Sigma_k) V^{\star\top}$, we have

$$M - M_\star = U_\star \, \mathrm{diag}(r_1, \ldots, r_d) \, V^{\star\top},$$

with $r_i$ defined in (32). Substituting these expressions into (34) and using $U^{\star\top} U_\star = I$, $V^{\star\top} V_\star = I$ yields

$$\nabla_{W_{h\ell}} \mathcal{L} = Q_{h,\ell+1}\Big( \prod_{j=\ell+1}^{L_h} \Sigma_{h,j} \Big) \mathrm{diag}(r_1, \ldots, r_d) \Big( \prod_{j=1}^{\ell-1} \Sigma_{h,j} \Big) Q_{h,\ell}^{\top}.$$

All matrices in the middle are diagonal and commute, hence the gradient shares the same left/right orthogonals as $W_{h\ell}$ and has diagonal entries

$$\Big[ \nabla_{W_{h\ell}} \mathcal{L} \Big]_{i,i} = r_i \prod_{j \neq \ell} \sigma_{h,j,i}.$$

Therefore the GD update preserves the SVS chart and, at the level of diagonal entries, is exactly (33).

Now assume $\Theta(t) \in \mathcal{B}$. Then for each fixed $(h, i)$ all layer singular values are equal: $\sigma_{h,1,i}(t) = \cdots = \sigma_{h,L_h,i}(t) =: s_{hi}(t)$. Hence the multiplicative factor in (33) becomes

$$\prod_{j \neq \ell} \sigma_{h,j,i}(t) = s_{hi}(t)^{L_h - 1},$$

which is independent of $\ell$. Thus every layer in the same pathway receives the same update in mode $i$, implying $\sigma_{h,1,i}(t+1) = \cdots = \sigma_{h,L_h,i}(t+1)$. This is precisely $\Theta(t+1) \in \mathcal{B}$. The final claim follows by induction on $t$. $\qquad\square$

**Implication for our initialization.** Our depth-balanced diagonal initialization (Eq. (11)) satisfies $\sigma_{h,1,i}(0) = \cdots = \sigma_{h,L_h,i}(0)$ for each pathway $h$ and mode $i$. Hence, *conditional on remaining in the SVS set*, Proposition B.4 justifies restricting the analysis to the depth-balanced manifold throughout training.

## C. Sharpness Reduction in Multi-Pathway Linear Models

In this section, we rigorously derive the relationship between the number of parallel pathways $H$ and the loss sharpness at a balanced global minimum. We demonstrate that distributing a target feature across multiple pathways significantly reduces the network's sharpness, thereby extending the edge of stability regime.

### C.1. Problem Formulation and Hessian Structure

Consider a Deep Linear Network (DLN) composed of $H$ parallel pathways, where the output matrix $M$ is the sum of the outputs of individual pathways $\Omega_h$. The objective is to minimize the squared Frobenius loss:

$$\mathcal{L}(\Theta) = \frac{1}{2}\|M - M_\star\|_F^2 = \frac{1}{2}\left\|\sum_{h=1}^{H}\Omega_h(\Theta_h) - M_\star\right\|_F^2, \tag{35}$$

where $\Theta_h = (W_{h,1}, \ldots, W_{h,L})$ represents the parameters of the $h$-th pathway, and $\Omega_h = W_{h,L}\cdots W_{h,1}$.

To analyze the sharpness, we examine the Hessian of the loss, $\nabla^2\mathcal{L}(\Theta)$, at a global minima where $M = M_\star$. Utilizing the vectorization of parameters, the Hessian can be decomposed into blocks corresponding to the interactions between pathways $h$ and $k$. Following the derivation of Ghosh et al. (2025) in Appendix C.3.2, the $(h, k)$-th block of the Hessian is given by:

$$\mathbf{H}_{h,k} = \nabla_{\mathrm{vec}(\Theta_h)}\nabla_{\mathrm{vec}(\Theta_k)}^{\top}\mathcal{L} \approx \mathbf{A}_h^{\top}\mathbf{A}_k, \tag{36}$$

### C.2. Mode-Wise Decoupling in the SVS Set

We now refine the argument by leveraging the *singular vector stationary* (SVS) structure used in Ghosh et al. (2025). Assume that training is restricted to the SVS set associated with the target SVD

$$M_\star = U\,\Sigma_\star\,V^{\top}, \qquad \Sigma_\star = \mathrm{diag}(\sigma_{\star 1}, \ldots, \sigma_{\star d}),$$

so that the singular vectors of all layer matrices remain fixed (equal to those of $M_\star$) throughout training. In particular, for each pathway $h$ and layer $\ell$ we may write

$$W_{h\ell} = U\,\Sigma_{h\ell}\,V^{\top}, \qquad \Sigma_{h\ell} = \mathrm{diag}(\sigma_{h\ell,1}, \ldots, \sigma_{h\ell,d}),$$

hence each end-to-end map is

$$\Omega_h = U\,\Sigma_h\,V^{\top},$$
$$\Sigma_h = \prod_{\ell=1}^{L}\Sigma_{h\ell} = \mathrm{diag}(\sigma_{h1}, \ldots, \sigma_{hd}),$$
$$\sigma_{hi} := \prod_{\ell=1}^{L}\sigma_{hi}^{(\ell)}.$$

Since $M = \sum_{h=1}^{H}\Omega_h$ and all $\Omega_h$ share the same singular vectors $(U, V)$, the total map is also diagonal in the same basis:

$$M = U\left(\sum_{h=1}^{H}\Sigma_h\right)V^{\top}$$
$$\implies \sigma_i(M) = \sum_{h=1}^{H}\sigma_{hi} \quad \text{for each mode } i \in [d].$$

Therefore the loss decomposes into a sum of independent *mode losses*

$$\mathcal{L}(\Theta) = \frac{1}{2} \|M - M_\star\|_F^2 = \frac{1}{2} \sum_{i=1}^{d} \left( \sum_{h=1}^{H} \sigma_{hi} - \sigma_{\star i} \right)^2 =: \sum_{i=1}^{d} \mathcal{L}_i. \tag{37}$$

This SVS decoupling implies that the Hessian decomposes (up to permutation) into blocks indexed by modes $i$, and the dominant positive-curvature directions can be analyzed mode-by-mode.

### C.3. Mode-$i$ Hessian Eigenvalue on the Depth-Balanced Manifold

We refine the Hessian expression starting from

$$\mathbf{H}_{h,k} = \nabla_{\mathrm{vec}(\Theta_h)} \nabla_{\mathrm{vec}(\Theta_k)}^{\top} \mathcal{L} \approx A_h^{\top} A_k$$

at a global minimum ($M = M_\star$), where $A_h$ is the Jacobian of $\mathrm{vec}(\Omega_h)$ w.r.t. $\mathrm{vec}(\Theta_h)$. Inside the SVS set, the Jacobian further decomposes across modes. Concretely, the scalar residual for mode $i$ is

$$r_i := \sum_{h=1}^{H} \sigma_{hi} - \sigma_{\star i},$$

so $\mathcal{L}_i = \frac{1}{2} r_i^2$ and, at any global minimum, $r_i = 0$ for all $i$. Hence the mode-$i$ Hessian is a Gram matrix of the mode-$i$ Jacobian:

$$\nabla^2 \mathcal{L}_i \big|_{r_i=0} = J_i^{\top} J_i.$$

**Depth-wise balance.** We now specialize to the *depth-balanced* manifold within each pathway (consistent with balanced initialization and the "balanced minimum" analyzed in Ghosh et al. (2025)):

$$\sigma_{hi}^{(\ell)} = \sigma_{hi}^{1/L} \qquad \forall\, h \in [H],\ \ell \in [L],\ i \in [d]. \tag{38}$$

That is, depth-wise the singular value for a fixed mode $i$ is equally split across layers, while *pathway-wise* the end-to-end values $\sigma_{hi}$ may be unbalanced across $h$.

**Jacobian in mode $i$.** Consider the coordinates $\{\sigma_{hi}^{(\ell)}\}_{h,\ell}$ in mode $i$. Since $\sigma_{hi} = \prod_{\ell=1}^{L} \sigma_{hi}^{(\ell)}$, we have

$$\frac{\partial \sigma_{hi}}{\partial \sigma_{hi}^{(\ell)}} = \frac{\sigma_{hi}}{\sigma_{hi}^{(\ell)}} \overset{(38)}{=} \frac{\sigma_{hi}}{\sigma_{hi}^{1/L}} = \sigma_{hi}^{1-\frac{1}{L}}.$$

Because $r_i = \sum_h \sigma_{hi} - \sigma_{\star i}$, the mode-$i$ Jacobian entries are

$$\frac{\partial r_i}{\partial \sigma_{hi}^{(\ell)}} = \frac{\partial \sigma_{hi}}{\partial \sigma_{hi}^{(\ell)}} = \sigma_{hi}^{1-\frac{1}{L}}.$$

Thus, restricted to these depth-balanced SVS coordinates, the mode-$i$ Jacobian vector is

$$J_i = \Big( \underbrace{\sigma_{1i}^{1-1/L}, \ldots, \sigma_{1i}^{1-1/L}}_{L \text{ entries}}, \underbrace{\sigma_{2i}^{1-1/L}, \ldots, \sigma_{2i}^{1-1/L}}_{L \text{ entries}}, \ldots, \underbrace{\sigma_{Hi}^{1-1/L}, \ldots, \sigma_{Hi}^{1-1/L}}_{L \text{ entries}} \Big),$$

and therefore its squared norm is

$$\|J_i\|_2^2 = \sum_{h=1}^{H} \sum_{\ell=1}^{L} \left( \sigma_{hi}^{1-1/L} \right)^2 = L \sum_{h=1}^{H} \sigma_{hi}^{2-\frac{2}{L}}. \tag{39}$$

**Lemma C.1** (Mode-wise dominant eigenvalue). *At any global minimum within the SVS set, and restricted to the depth-balanced manifold* (38)*, the Hessian contribution from mode $i$ has a single nonzero eigenvalue given by*

$$\lambda_i \; = \; L \sum_{h=1}^{H} \sigma_{hi}^{2-\frac{2}{L}}, \tag{40}$$

*with eigenvector proportional to the mode-$i$ Jacobian direction $J_i$.*

*Proof.* At a global minimum $r_i = 0$, we have $\nabla^2 \mathcal{L}_i = J_i^\top J_i$, which is rank-one on the considered coordinates. Hence its only nonzero eigenvalue equals $\|J_i\|_2^2$, which is (39). □

### C.4. Pathway-Wise Balancing Minimizes Sharpness in Each Mode

Lemma C.1 shows that, for a fixed mode $i$, the sharpness contribution depends on the pathway end-to-end singular values $\{\sigma_{hi}\}_{h=1}^{H}$ through the convex functional $\sum_h \sigma_{hi}^p$ with

$$p \; := \; 2 - \frac{2}{L}.$$

For $L > 2$, we have $p > 1$, so $x \mapsto x^p$ is convex on $\mathbb{R}_+$.

At a global minimum, mode $i$ must satisfy the constraint (from (37))

$$\sum_{h=1}^{H} \sigma_{hi} \; = \; \sigma_{\star i}. \tag{41}$$

We now optimize (40) subject to (41).

**Theorem C.2** (Pathway-wise equal split minimizes mode-$i$ sharpness). *Assume $L > 2$ (so $p > 1$). Among all global minima satisfying* (41)*, the mode-$i$ eigenvalue* (40) *is minimized when the pathway contributions are equal:*

$$\sigma_{1i} = \cdots = \sigma_{Hi} = \frac{\sigma_{\star i}}{H}.$$

*In that case,*

$$\lambda_i^{\min} \; = \; L H^{\frac{2}{L}-1} \sigma_{\star i}^{2-\frac{2}{L}}. \tag{42}$$

*Proof.* By Jensen's inequality for the convex function $x^p$ (with $p > 1$),

$$\frac{1}{H} \sum_{h=1}^{H} \sigma_{hi}^p \; \geq \; \left( \frac{1}{H} \sum_{h=1}^{H} \sigma_{hi} \right)^p = \left( \frac{\sigma_{\star i}}{H} \right)^p,$$

where we used the constraint (41). Multiplying by $LH$ yields

$$\lambda_i = L \sum_{h=1}^{H} \sigma_{hi}^p \; \geq \; LH \left( \frac{\sigma_{\star i}}{H} \right)^p,$$

with equality iff $\sigma_{1i} = \cdots = \sigma_{Hi} = \sigma_{\star i}/H$. Substituting $p = 2 - \frac{2}{L}$ gives (42). □

**Improvement over a "single-path" allocation.** Consider the extreme (maximally unbalanced) allocation that places all of mode $i$ into one pathway:

$$\sigma_{1i} = \sigma_{\star i}, \qquad \sigma_{2i} = \cdots = \sigma_{Hi} = 0.$$

Then (40) gives

$$\lambda_i^{\text{one path}} = L \sigma_{\star i}^{2-\frac{2}{L}}.$$

Comparing with the minimum (42), we obtain the reduction factor

$$\frac{\lambda_i^{\min}}{\lambda_i^{\text{one path}}} = \frac{L\,H\left(\frac{\sigma_{\star i}}{H}\right)^{2-\frac{2}{L}}}{L\,\sigma_{\star i}^{2-\frac{2}{L}}} = H^{\frac{2}{L}-1}. \tag{43}$$

Thus, for $L \geq 3$, distributing a mode evenly across $H$ pathways strictly reduces its curvature by a factor $(H)^{\frac{2}{L}-1}$ relative to concentrating it in a single pathway.

### C.5. From Mode-wise Eigenvalues to Global Sharpness

The full Hessian (restricted to the SVS) decomposes into independent mode contributions. Each mode $i$ contributes a dominant positive eigenvalue $\lambda_i$ given by (40), while additional eigenvalues correspond to symmetry/gauge directions (often yielding zero eigenvalues) and to non-dominant curvature within each mode. Consequently, the sharpness satisfies

$$\lambda_{\max} = \max_{i \in [d]} \lambda_i.$$

Under the common ordering $\sigma_{\star 1} \geq \sigma_{\star 2} \geq \cdots \geq \sigma_{\star d}$, the minimized mode-wise eigenvalues (42) are monotone in $\sigma_{\star i}$, hence the largest eigenvalue is achieved by the top target mode:

$$\lambda_{\max} = \lambda_1^{\min} = L\,H^{\frac{2}{L}-1}\,\sigma_{\star 1}^{2-\frac{2}{L}}. \tag{44}$$

Moreover, the remaining dominant eigenvalues align with the remaining target modes: for each $i$, the curvature scale is set by $\sigma_{\star i}$ through (42). In particular, when $L > 2$, increasing the number of pathways $H$ reduces $\lambda_{\max}$ as $H^{\frac{2}{L}-1}$, thereby extending the edge-of-stability regime to larger learning rates.

### C.6. Proof of Proposition 5.1

We extend the mode-wise SVS analysis to the case where pathway $h$ has (possibly different) depth $L_h$. Recall that, in mode $i$, each pathway contributes an end-to-end singular value

$$\sigma_{hi} = \prod_{\ell=1}^{L_h} \sigma_{hi}^{(\ell)},$$

and the mode-$i$ residual and loss are

$$r_i := \sum_{h=1}^{H} \sigma_{hi} - \sigma_{\star i}, \qquad \mathcal{L}_i = \tfrac{1}{2}r_i^2.$$

At any global minimum we have $r_i = 0$, hence (as in the homogeneous-depth case)

$$\nabla^2 \mathcal{L}_i\big|_{r_i=0} = J_i^\top J_i,$$

where $J_i$ denotes the Jacobian of $r_i$ with respect to the mode-$i$ coordinates.

**Depth-balanced manifold with heterogeneous depth.** Assume we restrict to the depth-balanced manifold *within each pathway*:

$$\sigma_{hi}^{(\ell)} = \sigma_{hi}^{1/L_h} \qquad \forall\, h \in [H],\ \ell \in [L_h],\ i \in [d]. \tag{45}$$

That is, for fixed $(h, i)$, the contribution of mode $i$ is evenly split across the $L_h$ layers of pathway $h$.

**Mode-$i$ Jacobian entries.** Fix a pathway $h$. Since $\sigma_{hi} = \prod_{\ell=1}^{L_h} \sigma_{hi}^{(\ell)}$, we have for each $\ell \in [L_h]$

$$\frac{\partial \sigma_{hi}}{\partial \sigma_{hi}^{(\ell)}} = \frac{\sigma_{hi}}{\sigma_{hi}^{(\ell)}} \overset{(45)}{=} \frac{\sigma_{hi}}{\sigma_{hi}^{1/L_h}} = \sigma_{hi}^{1-\frac{1}{L_h}}.$$

Because $r_i = \sum_{h'} \sigma_{h'i} - \sigma_{\star i}$ depends on $\sigma_{hi}$ only through this additive term, it follows that

$$\frac{\partial r_i}{\partial \sigma_{hi}^{(\ell)}} = \frac{\partial \sigma_{hi}}{\partial \sigma_{hi}^{(\ell)}} = \sigma_{hi}^{1-\frac{1}{L_h}} \qquad \forall\, \ell \in [L_h].$$

Therefore, in the coordinate system $\{\sigma_{hi}^{(\ell)}\}_{h,\ell}$, the mode-$i$ Jacobian vector $J_i$ consists of $L_h$ repeated entries of magnitude $\sigma_{hi}^{1-1/L_h}$ for each pathway $h$.

**Rank-one Hessian and its dominant eigenvalue.** Since $\nabla^2 \mathcal{L}_i|_{r_i=0} = J_i^\top J_i$ is a Gram matrix of a single vector, it is rank-one on these coordinates, with the unique nonzero eigenvalue equal to $\|J_i\|_2^2$. Using the structure above,

$$\|J_i\|_2^2 = \sum_{h=1}^{H} \sum_{\ell=1}^{L_h} \left( \sigma_{hi}^{1-\frac{1}{L_h}} \right)^2 = \sum_{h=1}^{H} L_h \, \sigma_{hi}^{2-\frac{2}{L_h}}.$$

Hence, on the depth-balanced manifold, the dominant (and only nonzero) eigenvalue contributed by mode $i$ is

$$\lambda_i \;=\; \sum_{h=1}^{H} L_h \, \sigma_{hi}^{2-\frac{2}{L_h}},$$

with eigenvector proportional to $J_i$. This proves Proposition 5.1.

## D. Stability Constraints Imply Drift Toward Pathway-Balanced Minima

**Background: self-stabilization as constrained minimization.** Self-stabilization theory (Damian et al., 2023) argues that, near the edge of stability, gradient descent (GD) implicitly tracks a projected GD trajectory that solves a sharpness-constrained problem:

$$\min_{\Theta} \; \mathcal{L}(\Theta) \quad \text{s.t.} \quad S(\Theta) \leq \frac{2}{\eta}, \tag{46}$$

where $S(\Theta) = \lambda_{\max}(\nabla^2 \mathcal{L}(\Theta))$ is the sharpness and $\eta$ is the step size. (See Damian et al. (2023) for a formal projected-dynamics statement.)

**Our setting.** We consider the SVS regime, where all layers share the singular vectors of the target $M_\star = V \Sigma_\star V^\top$, so the dynamics decouple across modes $i \in [d]$ and

$$\mathcal{L}(\Theta) = \frac{1}{2} \sum_{i=1}^{d} \left( \sum_{h=1}^{H} \sigma_{hi} - \sigma_{\star i} \right)^2 =: \sum_{i=1}^{d} \mathcal{L}_i. \tag{47}$$

We also restrict to the depth-balanced manifold within each pathway (homogeneous depth $L$): $\sigma_{hi}^{(\ell)} = \sigma_{hi}^{1/L}$ for all $h, \ell, i$.

### D.1. Local GD stability at a global minimum equals a sharpness bound

Let $\Theta_\star$ be any global minimum within SVS and depth-balance, so $\sum_h \sigma_{hi} = \sigma_{\star i}$ for all $i$. The mode-$i$ Hessian contribution is rank-one on the SVS depth-balanced coordinates, with the single nonzero eigenvalue

$$\lambda_i(\Theta_\star) = L \sum_{h=1}^{H} \sigma_{hi}^{2-\frac{2}{L}}. \tag{48}$$

(Equivalently, $\nabla^2 \mathcal{L}_i(\Theta_\star) = J_i^\top J_i$ with $\|J_i\|_2^2 = \lambda_i$.)

**Proposition D.1** (GD stability $\Leftrightarrow$ sharpness constraint). *Consider full-batch GD $\Theta_{t+1} = \Theta_t - \eta \nabla \mathcal{L}(\Theta_t)$. At a global minimum $\Theta_\star$ (SVS + depth-balance), the linearized GD map has eigenvalues $1 - \eta \lambda_i(\Theta_\star)$ on each mode-$i$ curvature direction and eigenvalue 1 on the flat directions. Hence $\Theta_\star$ is (linearly) stable iff*

$$\eta \, \lambda_{\max}(\Theta_\star) < 2, \qquad \lambda_{\max}(\Theta_\star) := \max_i \lambda_i(\Theta_\star), \tag{49}$$

*i.e., iff $S(\Theta_\star) \leq 2/\eta$.*

*Proof.* Linearizing GD at $\Theta_\star$ yields $\delta\Theta_{t+1} = (I - \eta \nabla^2 \mathcal{L}(\Theta_\star))\delta\Theta_t$. In SVS + depth-balance, the Hessian is block-diagonal across modes, and each mode-$i$ block is rank-one with eigenvalue $\lambda_i$ in (48). Stability on a positive-curvature direction requires $|1 - \eta\lambda_i| < 1$, i.e., $0 < \eta\lambda_i < 2$. Taking the maximum over $i$ gives (49). $\square$

### D.2. Why the stability constraint selects pathway-balanced minima

For fixed mode $i$ and homogeneous depth $L > 2$, define $p := 2 - \frac{2}{L} > 1$. At any global minimum, the pathway end-to-end singular values satisfy the conservation constraint $\sum_{h=1}^{H} \sigma_{hi} = \sigma_{\star i}$, while the mode-$i$ sharpness is proportional to $\sum_h \sigma_{hi}^p$ by (48).

**Theorem D.2** (Balanced split is the flattest global minimum in each mode)**.** *Assume $L > 2$. Among all global minima satisfying $\sum_h \sigma_{hi} = \sigma_{\star i}$, the mode-$i$ sharpness $\lambda_i$ is minimized uniquely by the equal split*

$$\sigma_{1i} = \cdots = \sigma_{Hi} = \frac{\sigma_{\star i}}{H}, \qquad \lambda_i^{\min} = L \, H^{\frac{2}{L}-1} \, (\sigma_{\star i})^{2-\frac{2}{L}}. \tag{50}$$

*Proof.* Since $p > 1$, $x \mapsto x^p$ is convex on $\mathbb{R}_+$. By Jensen, $\frac{1}{H} \sum_h \sigma_{hi}^p \geq \left( \frac{1}{H} \sum_h \sigma_{hi} \right)^p = (\sigma_{\star i}/H)^p$, with equality iff all $\sigma_{hi}$ are equal. Multiply both sides by $LH$ and substitute $p = 2 - 2/L$. $\qquad\square$

**Consequence (stability-driven drift).** Combine Proposition D.1 and Theorem D.2. If the stepsize satisfies

$$\frac{2}{\lambda_i^{\text{one-path}}} < \eta < \frac{2}{\lambda_i^{\min}}, \qquad \lambda_i^{\text{one-path}} := L(\sigma_{\star i})^{2-\frac{2}{L}}, \tag{51}$$

then any sharp "winner-takes-all" minimum (single-path allocation) is unstable, while the equal-split minimum is stable. Therefore, once GD enters a neighborhood where mode $i$ dominates the curvature, stability constraints alone rule out convergence to sparse allocations and select balanced allocations as the only stable zero-loss solutions. This provides a direct (assumption-free) mechanism linking "stability constraints" to drift toward pathway-balanced, flatter minima in our multi-path DLN.

## E. Decay of a Pathway Balancing Gap at the Edge of Stability

This appendix provides a local mechanism explaining why large-step GD reduces pathway imbalance near the balanced minimum. The analysis is carried out on the SVS depth-balanced manifold, where each singular mode decouples and the multi-pathway dynamics reduce to scalar recursions for the per-layer variables $a_{hi}(t)$. We first establish the invariance of this reduction and introduce gradient-flow leaf coordinates that separate the balanced direction from the pathway-imbalance directions. In these coordinates, the balanced minimum is the unique local minimizer of terminal sharpness, and deviations from the balanced leaf increase sharpness quadratically.

The main part of the appendix analyzes how discrete GD moves across these gradient-flow leaves in the Edge-of-Stability regime. We begin with a sign-free one-step contraction result in a strong-safety neighborhood. We then prove the central two-step result: under a local self-stabilization condition stating that the sharpness remains bracketed near the stability threshold, the residual normal form implies sign alternation and yields a two-step contraction of the pathway balancing gap, with an explicit quadratic correction. Thus sign alternation is not imposed as an independent hypothesis; it follows from the local EoS normal form together with sharpness bracketing. Finally, we translate the two-step contraction into a formal slow-time central-flow description, where the drift is active only when the sharpness excess $(\eta\lambda - 2)_+$ is positive, and we state precisely which conclusions are rigorous local statements and which remain formal asymptotics.

### E.1. Invariance of SVS and preservation of depth-wise balance

We extend the SVS invariance argument of Ghosh et al. (2025) (see also their deferred SVS proofs) to the multi-pathway architecture $M = \sum_{h=1}^{H} \Omega_h$ with $\Omega_h = \prod_{\ell=1}^{L} W_{h\ell}$.

**Lemma E.1** (SVS is invariant under diagonal (SVS) initialization)**.** *Assume the target is $M_\star = V\Sigma_\star V^\top$ and the initialization satisfies $W_{h\ell}(0) = V\Sigma_{h\ell}(0)V^\top$ with all $\Sigma_{h\ell}(0)$ diagonal (in particular, scalar multiples of the identity, as in our initialization). Then for all GD iterates $t \geq 0$,*

$$W_{h\ell}(t) = V\Sigma_{h\ell}(t)V^\top \quad \text{with} \quad \Sigma_{h\ell}(t) \text{ diagonal}, \tag{52}$$

*so the dynamics remain in the SVS set and decouple on singular values.*

*Proof.* Assume inductively that $W_{h\ell}(t) = V\Sigma_{h\ell}(t)V^\top$ with diagonal $\Sigma_{h\ell}(t)$ for all $h, \ell$. Each pathway product is $\Omega_h(t) = V\left(\prod_\ell \Sigma_{h\ell}(t)\right)V^\top$, hence the total map

$$M(t) = \sum_{h=1}^{H} \Omega_h(t) = V\left(\sum_{h=1}^{H}\prod_{\ell=1}^{L}\Sigma_{h\ell}(t)\right)V^\top$$

is diagonal in the same basis. The residual $R(t) := M(t) - M_\star$ therefore satisfies $R(t) = V\Delta(t)V^\top$ with diagonal $\Delta(t)$. For the squared Frobenius loss,

$$\nabla_{W_{h\ell}}\mathcal{L}(\Theta(t)) = W_{h,L:\ell+1}(t)^\top R(t) W_{h,\ell-1:1}(t)^\top.$$

Each factor is diagonal in the $V$ basis, so $\nabla_{W_{h\ell}}\mathcal{L} = VG_{h\ell}V^\top$ with diagonal $G_{h\ell}$. The GD update preserves the diagonal SVS form. $\square$

**Lemma E.2** (Depth-wise balance within each pathway is preserved). *Assume homogeneous depth $L$ and $\sigma_{hi}^{(1)}(t) = \cdots = \sigma_{hi}^{(L)}(t)$ for every $h, i$ at some iterate $t$. Then the same equalities hold at $t + 1$.*

*Proof.* By Lemma E.1, the dynamics reduce to diagonal singular values. Fix a mode $i$ and let $x_{h\ell} := \sigma_{hi}^{(\ell)}(t)$, $s_h := \prod_\ell x_{h\ell}$. The mode-$i$ residual is $r := \sum_h s_h - \sigma_{\star i}$ and $\mathcal{L}_i = \frac{1}{2}r^2$. By the product rule, $\partial\mathcal{L}_i/\partial x_{h\ell} = r \cdot s_h/x_{h\ell} = r\prod_{j\neq\ell} x_{hj}$. If $x_{h1} = \cdots = x_{hL}$, then $\prod_{j\neq\ell} x_{hj}$ is independent of $\ell$, so every layer in pathway $h$ receives the identical mode-$i$ update. Depth-wise balance is preserved. $\square$

### E.2. Mode-wise scalar dynamics on the depth-balanced manifold

By Lemma E.2, for each mode $i$ and pathway $h$ we can write $\sigma_{hi}^{(\ell)}(t) = a_{hi}(t)$ for all $\ell \in [L]$, so the end-to-end pathway singular value is $\sigma_{hi}(t) = a_{hi}(t)^L$. Let

$$r_i(t) := \sum_{h=1}^{H} a_{hi}(t)^L - \sigma_{\star i}. \tag{53}$$

Then GD on $\{a_{hi}\}$ obeys the exact recursion

$$a_{hi}(t+1) = a_{hi}(t) - \eta\, r_i(t)\, a_{hi}(t)^{L-1}. \tag{54}$$

**Notation for the remainder of this appendix.** We fix a mode $i$ and drop the index. Write $a_h := a_{hi}$, $r := r_i$, and

$$a_\star := (\sigma_\star/H)^{1/L}, \qquad \bar{a}(t) := \frac{1}{H}\sum_h a_h(t), \qquad \delta_h(t) := a_h(t) - \bar{a}(t), \qquad \mathcal{G}(t) := \sum_h \delta_h(t)^2.$$

The balanced equilibrium is $a_h = a_\star$ for all $h$, at which $\mathcal{G} = 0$. The sharpness of the balanced minimum is $\lambda^{\min} = LH^{2/L-1}\sigma_\star^{2-2/L} = LHa_\star^{2L-2}$.

### E.3. Gradient-flow foliation and local leaf-sharpness expansion

The discrete dynamics (54) has a gradient-flow (GF) limit

$$\dot{a}_h = -r\, a_h^{L-1}.$$

In this subsection we identify the conserved quantities of this GF limit and use them to define leaf coordinates for pathway imbalance. These coordinates separate motion along a GF leaf from motion across leaves. We then show that, on the zero-loss manifold, the terminal sharpness increases quadratically with the leaf-coordinate norm near the balanced leaf. Thus the balanced leaf is the unique local sharpness minimizer among nearby GF leaves.

**Centered leaf coordinates.** Define

$$q_h(t) := a_h(t)^{2-L}, \qquad \bar{q}(t) := \frac{1}{H}\sum_h q_h(t), \qquad z_h(t) := q_h(t) - \bar{q}(t),$$

so $\sum_h z_h(t) = 0$ and $z(t) \in \mathbf{1}^\perp \subset \mathbb{R}^H$.

**Proposition E.3** (GF foliation in centered coordinates). *Let $L > 2$. Under the gradient flow $\dot{a}_h = -r\, a_h^{L-1}$,*

$$\frac{d}{dt}q_h(t) = (L-2)\, r(t) \quad \text{for all } h,$$

*so the centered vector $z(t)$ is conserved. The phase space $\mathbb{R}_{>0}^H$ is therefore foliated by affine level sets in $q$-coordinates: for each $z \in \mathbf{1}^\perp$, the GF leaf is*

$$q = m\,\mathbf{1} + z, \qquad m > -\min_h z_h, \tag{55}$$

*and its image in $a$-coordinates is the one-dimensional curve $a_h = (m + z_h)^{-1/(L-2)}$. The unique leaf containing the balanced minimum is $z = 0$.*

*Proof.* $\frac{d}{dt}q_h = (2-L)a_h^{1-L}\dot{a}_h = (2-L)a_h^{1-L}(-r\, a_h^{L-1}) = (L-2)r$, independent of $h$. Hence $z_h(t) = q_h(t) - \bar{q}(t)$ is constant. The lower bound on $m$ in (55) ensures $q_h > 0$, equivalently $a_h \in \mathbb{R}_{>0}$. $\qquad\square$

**Equivalence of $V$ and $\mathcal{G}$.** The quantity naturally controlled by the GF foliation is the variance of the centered leaf coordinate,

$$V(t) := \sum_h z_h(t)^2 = \sum_h \big(q_h(t) - \bar{q}(t)\big)^2. \tag{56}$$

The quantity used in the main text to measure pathway imbalance is

$$\mathcal{G}(t) := \sum_h \delta_h(t)^2, \qquad \delta_h(t) := a_h(t) - \bar{a}(t).$$

Near the balanced point these two quantities are locally equivalent. Indeed, if $a_h = a_\star + \delta_h$ with $\sum_h \delta_h = 0$, then Taylor expansion of $q_h = a_h^{2-L}$ around $a_\star$ gives

$$q_h - \bar{q} = (2-L)a_\star^{1-L}\delta_h + O(\varepsilon^2), \qquad \varepsilon := \max_h |\delta_h|.$$

Consequently,

$$V = (L-2)^2\, a_\star^{2-2L}\, \mathcal{G} + O(\mathcal{G}^{3/2}). \tag{57}$$

Thus contraction of the leaf-coordinate variance $V$ is equivalent, locally, to contraction of the pathway balancing gap $\mathcal{G}$ up to a positive multiplicative factor.

**Proposition E.4** (Local sharpness expansion across GF leaves). *Let $a(z) \in \mathcal{M} = \{a : \sum_h a_h^L = \sigma_\star\}$ be the zero-loss point on the GF leaf indexed by $z \in \mathbf{1}^\perp$, and define the restricted-Hessian sharpness*

$$\Lambda(z) := L\sum_h a_h(z)^{2L-2}.$$

*Then $z = 0$ corresponds to the balanced minimum $a_h = a_\star$ with $\Lambda(0) = \lambda^{\min} = LHa_\star^{2L-2}$, and*

$$\Lambda(z) = \lambda^{\min} + \frac{L(L-1)}{L-2}\, a_\star^{4L-6}\, \|z\|^2 + O(\|z\|^3). \tag{58}$$

*Equivalently, with $P$ denoting the orthogonal projection onto $\mathbf{1}^\perp$,*

$$H_{\text{leaf}} = \frac{2L(L-1)}{L-2}\, a_\star^{4L-6}\, P \succ 0 \quad \text{on } \mathbf{1}^\perp. \tag{59}$$

*Hence the balanced leaf $z = 0$ is the unique local minimizer of the terminal sharpness among nearby GF leaves.*

*Proof sketch.* Parametrize the leaf as $q_h = m + z_h$. The constraint $\sum_h a_h^L = \sigma_\star$ fixes $m = m(z)$ implicitly; at $z = 0$, $m = a_\star^{2-L}$ and $a_h = a_\star$ for all $h$. A second-order expansion of $\Lambda(z)$ using $a_h = (m(z) + z_h)^{-1/(L-2)}$ together with the implicit derivative of $m(z)$ yields (58). The first-order term in $z$ vanishes by the Jensen argument of Theorem 4.2. $\qquad\square$

Combining (58) with (57) gives, on the zero-loss manifold,

$$\lambda(a) = \lambda^{\min} + C_\lambda \mathcal{G} + O(\mathcal{G}^{3/2}), \qquad C_\lambda := L(L-1)(L-2)\, a_\star^{2L-4} > 0. \tag{60}$$

### E.4. Sign-free step-wise contraction (strong-safety regime)

The change of variable $\tilde{a}_h := a_h^{2-L}$ converts the GD update (54) into a coordinate-wise application of a single one-dimensional map.

**Lemma E.5** (One-dimensional reduction). *Set $\tilde{a}_h(t) := a_h(t)^{2-L}$ and $c(t) := \eta\, r(t)$. Then for every $h$,*

$$\tilde{a}_h(t+1) = F_{c(t)}\big(\tilde{a}_h(t)\big), \qquad F_c(u) := \frac{u^{L-1}}{(u-c)^{L-2}}, \tag{61}$$

*provided $u - c > 0$.*

*Proof.* From (54), $a_h(t+1) = a_h(t)(1 - \eta r(t)\, a_h(t)^{L-2}) = a_h(t)(1 - c(t)/\tilde{a}_h(t))$. Hence $\tilde{a}_h(t+1) = a_h(t+1)^{2-L} = \tilde{a}_h(t)\big(\tilde{a}_h(t)/(\tilde{a}_h(t) - c)\big)^{L-2}$, which simplifies to the claimed form. $\qquad\square$

The crucial point is that $F_{c(t)}$ does *not* depend on $h$. Hence $V(t+1)$ is determined entirely by the dispersion of $F_{c(t)}$ on $\{\tilde{a}_h(t)\}_{h=1}^H$.

**Lemma E.6** (Monotonicity and non-expansion of $F_c$). *Let $L > 2$ and suppose either (i) $c \le 0$, or (ii) $c > 0$ with $c \max_h a_h^{L-2} \le 1/(L-1)$. Then on $I := [\min_h \tilde{a}_h, \max_h \tilde{a}_h]$,*

$$0 \le F_c'(u) \le 1, \tag{62}$$

*with strict inequality $F_c'(u) < 1$ whenever $c \ne 0$.*

*Proof.* $F_c'(u) = u^{L-2}\big(u - (L-1)c\big)/(u-c)^{L-1}$. The denominator is positive since $u > c$. The factor $u - (L-1)c$ is positive when $c \le 0$, and when $c > 0$ it is positive iff $u \ge (L-1)c$. In terms of $a_h$, $u = a_h^{2-L} \ge (L-1)c$ rewrites as $\eta r\, a_h^{L-2} \le 1/(L-1)$, hypothesis (ii). For non-expansion, let $g(c) := (u-c)^{L-1} - u^{L-2}\big(u - (L-1)c\big)$. Then $g(0) = 0$ and $g'(c) = (L-1)\big[u^{L-2} - (u-c)^{L-2}\big]$ is strictly positive for $c > 0$ and strictly negative for $c < 0$, hence $g(c) > 0$ for all $c \ne 0$. This is equivalent to $F_c'(u) < 1$. $\qquad\square$

**Definition E.7** (Strong-safety condition). *At iterate $t$,*

$$r(t) \le 0, \qquad \text{or} \qquad r(t) > 0 \text{ and } \eta\, r(t)\, \max_h a_h(t)^{L-2} \le \frac{1}{L-1}. \tag{$\star$}$$

**Theorem E.8** (Sign-free step-wise contraction). *Let $L > 2$ and suppose the strong-safety condition $(\star)$ holds at iterate $t$. Then*

$$V(t+1) \le V(t), \tag{63}$$

*with strict inequality whenever $r(t) \ne 0$ and the $a_h(t)$ are not all equal.*

*Proof.* By Lemma E.5, $\tilde{a}_h(t+1) = F_{c(t)}(\tilde{a}_h(t))$ for every $h$. By Lemma E.6, $F_{c(t)}$ has derivative in $[0,1]$ on the relevant interval, with $F_{c(t)}' < 1$ strictly whenever $c(t) \ne 0$. The mean-value theorem gives $|F_{c(t)}(\tilde{a}_h) - F_{c(t)}(\tilde{a}_k)| \le |\tilde{a}_h - \tilde{a}_k|$, strict unless $\tilde{a}_h = \tilde{a}_k$. The variance $V$ is proportional to $\sum_{h,k}(\tilde{a}_h - \tilde{a}_k)^2$, so summing pairwise inequalities yields $V(t+1) \le V(t)$ with the stated strictness. $\qquad\square$

**Scope of Theorem E.8.** The strong-safety condition $(\star)$ is strictly more restrictive than the linear stability bound $\eta\lambda^{\min} < 2$ characterizing the basin of the balanced minimum. In particular, large-amplitude EoS oscillations generally violate $(\star)$. Theorem E.8 therefore guarantees step-wise monotonicity only inside an inner neighborhood of the balanced minimum, where the strong-safety condition is automatically satisfied (since $r \to 0$ implies $c \to 0$). To handle the EoS regime itself, we pass to two-step contraction with explicit quadratic correction; this is the content of the next subsection.

### E.5. Two-step contraction from the EoS normal form

The one-step contraction theorem above applies only in the strong-safety regime. To analyze the actual Edge-of-Stability regime, where the residual oscillates and the strong-safety condition may fail, we pass to a two-step description. The key input is a local normal form for the scalar residual recursion, retained one order beyond the linear EoS approximation. This normal form has two consequences: first, sign alternation of the residual follows from sharpness bracketing near the stability threshold; second, the quadratic term in the residual recursion contributes a positive correction to the two-step contraction coefficient.

**Lemma E.9** (Residual recursion: EoS normal form). *Let* $\lambda(\Theta_t) := L \sum_h a_h(t)^{2L-2}$, *define* $\Lambda_3(\Theta_t) := L(L-1) \sum_h a_h(t)^{3L-4}$ *and* $\Lambda_4(\Theta_t) := L(L-1)(L-2) \sum_h a_h(t)^{4L-6}$, *and set*

$$\beta(t) := 2 - \eta\lambda(\Theta_t), \quad A(t) := \tfrac{\eta^2}{2}\Lambda_3(\Theta_t) > 0, \quad B(t) := \tfrac{\eta^3}{6}\Lambda_4(\Theta_t) > 0.$$

*Then*

$$r(t+1) = -r(t) + \beta(t)\, r(t) + A(t)\, r(t)^2 - B(t)\, r(t)^3 + O\big(r(t)^4\big). \tag{64}$$

*Proof.* Expand $a_h(t+1)^L = a_h(t)^L \big(1 - \eta r(t)\, a_h(t)^{L-2}\big)^L$ using $(1-x)^L = 1 - Lx + \binom{L}{2}x^2 - \binom{L}{3}x^3 + O(x^4)$ and sum over $h$, identifying $\lambda, \Lambda_3, \Lambda_4$. $\square$

**Hypothesis E.10** (Self-stabilized EoS regime). There exist a neighborhood $\mathcal{N}$ of the balanced minimum, constants $\varepsilon_0, \rho_0, C_\beta > 0$, and a time $t_0$ such that for all $t \geq t_0$ with $\Theta_t \in \mathcal{N}$:

1. *Locality:* $\max_h |\delta_h(t)| \leq \varepsilon_0$ and $|r(t)| \leq \rho_0$;

2. *Sharpness bracketing:* $|\beta(t)| = |2 - \eta\lambda(\Theta_t)| \leq C_\beta\, r(t)^2$.

**Derived sign alternation.** Hypothesis E.10(ii), combined with Lemma E.9, gives

$$r(t+1) = -r(t) + O(r(t)^2).$$

Therefore, for all sufficiently small $r(t) \neq 0$, the residuals $r(t)$ and $r(t+1)$ have opposite signs. In particular, the sign alternation used in the two-step contraction is not imposed as a separate dynamical assumption; it is a local consequence of the EoS residual normal form together with sharpness bracketing.

**Two-step expansion.** Under Hypothesis E.10,

$$r(t) + r(t+1) = A(t)\, r(t)^2 + O\big(r(t)^3\big), \qquad r(t)r(t+1) = -r(t)^2 + O\big(r(t)^3\big). \tag{65}$$

The crucial point — and the source of the corrected contraction coefficient — is that $r(t) + r(t+1)$ is $O(r^2)$, *not* $O(r^3)$. The frozen-linearization two-step transverse multiplier at the balanced point is

$$m_t = (1 - \kappa_\star r(t))(1 - \kappa_\star r(t+1)) = 1 - \big(\kappa_\star^2 + \kappa_\star A(t)\big)r(t)^2 + O\big(r(t)^3\big),$$

where $\kappa_\star := \eta(L-1)a_\star^{L-2}$, hence $m_t^2 = 1 - 2(\kappa_\star^2 + \kappa_\star A(t))r(t)^2 + O(r(t)^3)$.

**Robust formulation.** For a fully local theorem, the step-to-step variation of $\kappa_t$ around $\kappa_\star$ should be absorbed into the local error term rather than claimed exactly. We state the robust form, which is the version we use.

**Theorem E.11** (Robust two-step contraction from the EoS normal form). *Let* $L > 2$. *Suppose Hypothesis E.10 holds at iterate $t$ and the iterates remain on the positive branch. Then there exist constants $c_0 > 0$ and $C > 0$, depending only on the local neighborhood, such that*

$$\mathcal{G}(t+2) \leq \Big(1 - c_0\, r(t)^2 + C\big(|r(t)|^3 + r(t)^2\varepsilon_t + r(t)^2\varepsilon_t^2\big)\Big) \mathcal{G}(t), \tag{66}$$

*where* $\varepsilon_t := \max_h |\delta_h(t)|$. *Equivalently, the remainder is $o(r(t)^2)$ relative to the leading term under the local scaling assumptions. Moreover, if the transverse linearization is frozen at the balanced point, the leading coefficient admits the explicit form*

$$c_0 = 2\big(\kappa_\star^2 + \kappa_\star A_\star\big), \qquad \kappa_\star = \eta(L-1)a_\star^{L-2}, \qquad A_\star = \tfrac{\eta^2}{2}L(L-1)H\, a_\star^{3L-4}.$$

*Whenever the error terms in (66) are dominated by $c_0\, r(t)^2$, the balancing gap contracts:* $\mathcal{G}(t+2) < \mathcal{G}(t)$.

*Proof sketch.* The relations (65) imply the linear part of the two-step transverse update has multiplier $1 - c\,r(t)^2 + O(r(t)^3 + r(t)^2\varepsilon_t)$ for some $c > 0$. Squaring gives the leading factor $1 - c_0\,r(t)^2$. The nonlinear transverse contributions from the quadratic Taylor correction of the per-pathway update are proportional to powers of $r(t)$ (since the GD map reduces to the identity at $r(t) = 0$), contributing $O\big((|r(t)|^3 + r(t)^2\varepsilon_t)\mathcal{G}(t)\big)$. This yields (66). $\qquad\square$

**Remark on the remainder.** The GD update $a_h^+ = a_h - \eta r\, a_h^{L-1}$ reduces to the identity when $r = 0$, so every term in the two-step error must carry at least one factor of $r(t)$. This is reflected in (66): there is no standalone $\varepsilon_t^4$ contribution. The compact form is $R(t) = o(r(t)^2\, \mathcal{G}(t))$ under the EoS scaling.

**Relation to the sign-alternating formulation.** A weaker way to state the two-step mechanism is to assume directly that the residual forms a small-amplitude two-cycle,

$$r(t+1) = -r(t) + O(r(t)^2).$$

Under this assumption, the leading part of Theorem E.11 reduces to the familiar contraction form

$$\mathcal{G}(t+2) \leq \big(1 - c\,r(t)^2\big)\mathcal{G}(t)$$

for some $c > 0$. The present formulation is stronger in two respects: the small-amplitude sign alternation is derived from the EoS sharpness bracketing, and the retained quadratic term in the residual normal form contributes the positive correction $\kappa_\star A_\star$ to the leading frozen contraction coefficient.

### E.6. Formal central-flow description of the balancing gap

We now translate the local two-step contraction into a formal slow-time ODE in the spirit of central flows (Cohen et al., 2025). This subsection should be read as a leading-order asymptotic description, not as a finite-time approximation theorem. Its purpose is to identify the direction of the slow cross-leaf drift and the stopping boundary selected by the EoS stability constraint.

The resulting picture distinguishes two cases. At the critical learning rate $\eta = 2/\lambda^{\min}$, the formal drift continues until full balance $\mathcal{G} = 0$. At intermediate learning rates, the drift stops earlier at the one-sided boundary where the sharpness reaches $2/\eta$; the prediction is then partial balance rather than exact balance.

**Sign convention.** Use the sharpness excess $\chi(\Theta) := \eta\lambda(\Theta) - 2$. The EoS regime is $\chi > 0$, where sustained residual oscillations occur; below the threshold, $\chi < 0$, the residual decays exponentially and the central-flow drift is absent.

**Self-stabilized variance.** The cubic restoring term in Lemma E.9 pins the time-averaged squared residual to scale with the positive part of $\chi$:

$$\langle r^2\rangle(\Theta) = K(\Theta)\,\chi(\Theta)_+ + O(\chi^2), \tag{67}$$

where $K(\Theta) > 0$ depends locally on $\Lambda_3, \Lambda_4$ via the standard self-stabilization calculation (Damian et al., 2023, Section 3). Crucially, the positive part $\chi_+$ appears: below the EoS threshold there is no oscillation and no drift.

**Theorem E.12** (Formal slow ODE for the balancing gap). *Let $L > 2$ and suppose Hypothesis E.10 holds along the trajectory. Define slow time $\tau := t/2$. To leading order in $\mathcal{G}$,*

$$\frac{d\mathcal{G}}{d\tau} = -K_0\left(\eta\lambda(\mathcal{G}) - 2\right)_+\mathcal{G} + O(\mathcal{G}^2), \tag{68}$$

*where $K_0 > 0$ collects the per-step contraction coefficient from Theorem E.11 and the self-stabilized residual scale, and $\lambda(\mathcal{G})$ is given by (60).*

**Stopping boundary, not stable equilibrium.** The right-hand side of (68) contains $\chi_+$, not $\chi$. Hence the drift acts only when $\lambda(\mathcal{G}) > 2/\eta$ and is *identically zero* when $\lambda(\mathcal{G}) \leq 2/\eta$. There is no restoring force in the opposite direction. The correct dynamical picture is therefore a *one-sided sliding boundary*: $\mathcal{G}$ decreases monotonically until $\lambda(\mathcal{G})$ reaches $2/\eta$, at which point the drift switches off and the iterate halts (at the slow-ODE level).

Combining with (60), the stopping condition is

$$\lambda(\mathcal{G}^\dagger) = \frac{2}{\eta}, \qquad \mathcal{G}^\dagger = \frac{2 - \eta\lambda^{\min}}{\eta\,L(L-1)(L-2)\,a_\star^{2L-4}} + O\big((\eta\lambda^{\min} - 2)^{3/2}\big). \tag{69}$$

**Three regimes.**

1. *Intermediate window* $2/S_1 < \eta < 2/\lambda^{\min}$. Here $\eta\lambda^{\min} < 2$, so the balanced point $\mathcal{G} = 0$ is below the EoS threshold ($\chi(0) < 0$). The drift is active when $\mathcal{G} > \mathcal{G}^\dagger > 0$ and inactive when $\mathcal{G} \leq \mathcal{G}^\dagger$. The formal ODE predicts *drift until the sharpness reaches $2/\eta$, then stop*; the resulting state is partially balanced, not fully balanced. This is a stopping boundary, not a two-sided stable equilibrium.

2. *Critical* $\eta = 2/\lambda^{\min}$. Here $\eta\lambda^{\min} = 2$, so $\chi(0) = 0$ and $\mathcal{G}^\dagger = 0$. The stopping boundary coincides with the balanced center, and the drift drives the trajectory all the way to $\mathcal{G} = 0$. At this learning rate the formal ODE selects the fully balanced minimum.

3. *Super-critical* $\eta > 2/\lambda^{\min}$. Here $\chi(0) > 0$: even the balanced minimum is GD-unstable in the local model. The slow ODE has no zero-loss stopping boundary in $\mathcal{G} \geq 0$; the trajectory must leave $\mathcal{N}$ or enter a different nonlinear regime, consistent with the worst-case-return analysis of Section 6.

**Recovery of the stability window (17).** The window $2/S_1 < \eta < 2/\lambda^{\min}$ of the main text is precisely the set of $\eta$ for which (a) the sharp single-path minimum is GD-unstable (so the symmetry-breaking phase cannot terminate there) and (b) the slow ODE admits a nontrivial stopping boundary $\mathcal{G}^\dagger \in [0, \mathcal{G}^{\text{1-path}})$. The endpoint $\eta = 2/\lambda^{\min}$ is the unique learning rate at which the stopping boundary collapses to the fully balanced minimum.

**Geometric interpretation.** Combining Theorem E.12 with Propositions E.3 and E.4: GD oscillations of amplitude $\langle r^2 \rangle \propto \chi_+$ produce a one-sided cross-leaf drift $d\mathcal{G}/d\tau \propto -\chi_+\mathcal{G}$. Via (57), the leaf coordinate $z$ tracks $\mathcal{G}$, so the trajectory drifts across GF leaves in the direction of decreasing $\|z\|$. The drift switches off precisely at the leaf whose terminal sharpness equals $2/\eta$ — i.e., at the boundary between the EoS regime and the linearly stable regime — and the trajectory stops there. At intermediate learning rates this stopping leaf is not $z = 0$ but a partially balanced leaf; at $\eta = 2/\lambda^{\min}$, the stopping leaf is exactly $z = 0$.

### E.7. Scope of the local analysis

We conclude by separating the rigorous local statements from the formal asymptotic interpretation.

1. **Rigorous reductions and local identities.** We prove SVS invariance and preservation of depth-wise balance (Lemmas E.1 and E.2), the exact scalar recursion $a_h^+ = a_h - \eta r\, a_h^{L-1}$ in (54), the GF foliation in centered coordinates $z \in \mathbf{1}^\perp$ (Proposition E.3), and the local sharpness expansion across GF leaves in (58). We also prove one-step contraction of the leaf-coordinate variance $V$ under the strong-safety condition (Theorem E.8) and the residual normal form (Lemma E.9).

2. **Local EoS contraction under self-stabilization.** The two-step contraction theorem (Theorem E.11) is a local statement near the balanced minimum, conditional on the self-stabilized EoS bracketing

$$|2 - \eta\lambda(\Theta_t)| = O(r(t)^2).$$

   Given this bracketing, the residual normal form implies sign alternation and the pathway balancing gap contracts over two steps whenever the local error terms are dominated by the leading quadratic contraction term. The bracketing condition itself is imported from the self-stabilization theory of GD at the EoS and is not proved from first principles here.

3. **Formal slow-time interpretation.** The central-flow equation (68) is a leading-order asymptotic description. It explains the direction of the slow drift and the one-sided stopping boundary selected by the stability threshold $\eta\lambda = 2$, but it is not a finite-time approximation theorem.

4. **Not proved globally.** We do not prove that every winner-takes-all transient enters the local self-stabilized EoS neighborhood in finite time. Establishing such a result would turn the local mechanism in this appendix into a full global re-balancing theorem. The experiments in Figures 1–4 support this global picture, but the present proof is local.

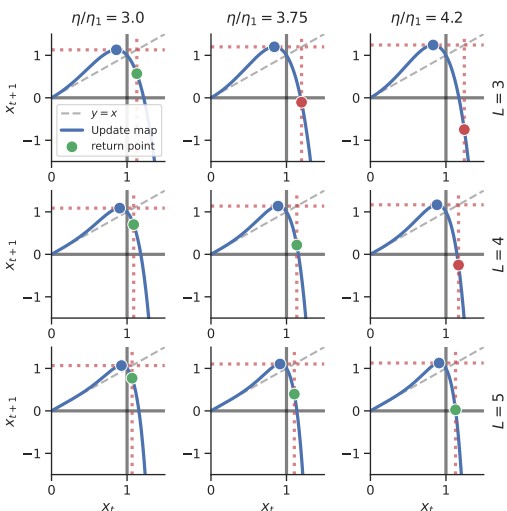

*Figure 6.* Deep linear chain update map (19) for different values of $\eta/\eta_1$ and $L$. The worst-case overshoot $(c, f_\eta(c))$ and the return point $(f_\eta(c), f_\eta(f_\eta(c)))$ illustrate how larger depth permits larger normalized steps before a sign flip occurs.

## F. Worst-Case Return Threshold for Deep Chains

This appendix provides the proofs and computational details for Section 6. Our goal is to characterize the largest learning rate for which the deep-chain GD map cannot cross the origin after the *worst* overshoot from the positive branch.

### F.1. From one-path concentration to a deep-chain map

On the SVS set restricted to the depth-balanced manifold, each pathway–mode pair admits a per-layer scalar $a_{h1}(t) \geq 0$ whose end-to-end singular value is $\sigma_{h1}(t) = a_{h1}(t)^L$. The corresponding mode-wise GD recursion is

$$a_{h1}(t+1) = a_{h1}(t) - \eta\, r_1(t)\, a_{h1}(t)^{L-1}, \tag{70}$$

$$r_1(t) = \sum_{k=1}^{H} a_{k1}(t)^L - \sigma_{\star 1}.$$

If, during symmetry breaking, the dominant mode is concentrated in one pathway $h = 1$ so that $a_{11}(t)^L \gg \sum_{k \neq 1} a_{k1}(t)^L$, then the residual satisfies $r_1(t) \approx a_{11}(t)^L - \sigma_{\star 1}$. Substituting this into (70) yields the single-variable deep-chain map

$$w_{t+1} = w_t - \eta\, w_t^{L-1}(w_t^L - \sigma_{\star 1}), \qquad w_t \equiv a_{11}(t),$$

which is (19).

### F.2. Normalization and geometric preliminaries

Let $w_\star = (\sigma_{\star 1})^{1/L}$ and define normalized variables

$$u := \frac{w}{w_\star} \in (0, \infty), \qquad \alpha := \eta\, (\sigma_{\star 1})^{2-2/L}. \tag{71}$$

Then (19) becomes the dimensionless map

$$u^+ = g_\alpha(u) := u - \alpha\, u^{L-1}(u^L - 1) = u + \alpha\, u^{L-1}(1 - u^L). \tag{72}$$

The overshoot-onset threshold $\eta_1 = 1/S_1$ corresponds to $\alpha = 1/L$, since $S_1 = L(\sigma_{\star 1})^{2-2/L}$.

**Lemma F.1** (Overshoot regime and monotonicity for $u \geq 1$). *Assume $\alpha > 1/L$. Then (i) there exist $u \in (0, 1)$ such that $g_\alpha(u) > 1$ (overshoot occurs), and (ii) $g_\alpha$ is strictly decreasing on $[1, \infty)$.*

*Proof.* We have $g_\alpha(1) = 1$ and

$$g'_\alpha(u) = 1 - \alpha\, u^{L-2}\Big((2L-1)u^L - (L-1)\Big), \qquad g'_\alpha(1) = 1 - \alpha L < 0 \quad (\alpha > 1/L),$$

so $g_\alpha$ exceeds 1 for $u < 1$ sufficiently close to 1, proving (i). For (ii), note that for every $u \geq 1$,

$$(2L-1)u^L - (L-1) \geq L, \qquad u^{L-2} \geq 1,$$

hence $g'_\alpha(u) \leq 1 - \alpha L < 0$ for all $u \geq 1$. $\qquad\square$

### F.3. Worst-case overshoot and the reduction to a single inequality

For $\alpha > 1/L$, define the worst-case overshoot value

$$v_{\max}(\alpha) := \max_{u \in [0,1]} g_\alpha(u). \tag{73}$$

By continuity, the maximum exists; by Lemma F.1(i), we have $v_{\max}(\alpha) > 1$ in the overshoot regime.

Before stating the reduction, write the two-step return property in normalized coordinates:

$$\forall\, u \in (0,1): \quad g_\alpha(u) > 1 \implies 0 < g_\alpha^{\circ 2}(u) < 1. \tag{74}$$

**Lemma F.2** (Two-step return reduces to positivity at $v_{\max}$). *Assume $\alpha > 1/L$. Then the WCR property (74) is equivalent to*

$$g_\alpha\big(v_{\max}(\alpha)\big) > 0. \tag{75}$$

*Proof.* Fix $\alpha > 1/L$ and let $u \in (0,1)$ be such that $g_\alpha(u) > 1$. By Lemma F.1(ii), $g_\alpha$ is strictly decreasing on $[1,\infty)$, hence

$$g_\alpha\big(g_\alpha(u)\big) \geq g_\alpha\big(v_{\max}(\alpha)\big).$$

Moreover, because $g_\alpha$ is decreasing on $[1,\infty)$ and $g_\alpha(1) = 1$, we automatically have $g_\alpha(g_\alpha(u)) < 1$ whenever $g_\alpha(u) > 1$. Therefore the only nontrivial part of two-step return is the lower bound $g_\alpha(g_\alpha(u)) > 0$, which is ensured for all overshooting $u$ iff the worst case is positive: $g_\alpha(v_{\max}(\alpha)) > 0$. $\qquad\square$

### F.4. Boundary system and the 1D root equation

Let $c \in (0,1)$ be any maximizer attaining $v_{\max}(\alpha) = g_\alpha(c)$; then necessarily $g'_\alpha(c) = 0$. Define

$$y := c^L, \qquad A(y) := (2L-2)y - (L-2), \qquad B(y) := (2L-1)y - (L-1).$$

A direct substitution yields the standard identities

$$g'_\alpha(c) = 0 \iff \alpha = \frac{1}{c^{L-2}\, B(y)} = \frac{y^{2/L-1}}{B(y)}, \tag{76}$$

$$v_{\max}(\alpha) = g_\alpha(c) = c\,\frac{A(y)}{B(y)}. \tag{77}$$

**Definition F.3** (Normalized WCR constant). Define

$$\alpha_{\mathrm{WCR}}(L) := \sup\{\alpha > 0: \ g_\alpha(v_{\max}(\alpha)) > 0\}. \tag{78}$$

Equivalently, $\alpha_{\mathrm{WCR}}(L)$ is the unique boundary value for which $g_\alpha(v_{\max}(\alpha)) = 0$.

**Lemma F.4** (Boundary system). *At $\alpha = \alpha_{\mathrm{WCR}}(L)$, there exists $c \in (0,1)$ such that*

$$g'_\alpha(c) = 0, \qquad g_\alpha\big(g_\alpha(c)\big) = 0. \tag{79}$$

*Conversely, any $(\alpha, c)$ satisfying (79) yields $\alpha = \alpha_{\mathrm{WCR}}(L)$.*

*Proof.* By Lemma F.2, the boundary is characterized by $g_\alpha(v_{\max}(\alpha)) = 0$. At the maximizer $c$ attaining $v_{\max}(\alpha) = g_\alpha(c)$ we have $g'_\alpha(c) = 0$, hence (79). Conversely, if (79) holds, then the worst overshoot lands exactly at the origin in the next step, so the corresponding $\alpha$ is the WCR boundary. □

Combining the boundary identities gives the one-dimensional root equation

$$B(y)^{2L-1} = A(y)^{L-2}\Big(y\,A(y)^L - B(y)^L\Big), \tag{80}$$

on the interval

$$y \in \Big(\frac{L-1}{2L-1}, 1\Big).$$

Once this root is found, the normalized WCR constant is

$$\gamma_{\mathrm{WCR}}(L) = L\,\frac{y^{2/L-1}}{B(y)}. \tag{81}$$

*Proof of Theorem 6.2.* Let $(\alpha, c)$ satisfy (79). Using (76) and (77), write $y = c^L$ and $v_{\max} = cA/B$. The condition $g_\alpha(v_{\max}) = 0$ implies

$$\alpha = \frac{1}{v_{\max}^{L-2}(v_{\max}^L - 1)}.$$

Combining this with (76) and substituting $v_{\max}^L = y(A/B)^L$ yields, after cancellation and algebra, (80). Finally, $\gamma_{\mathrm{WCR}}(L) = \eta_{\mathrm{WCR}}S_1 = L\alpha_{\mathrm{WCR}}(L)$ and $\alpha_{\mathrm{WCR}}(L) = y^{2/L-1}/B(y)$ by (76), proving (81) and (20). □

### F.5. Numerical computation (bisection) and representative values

Equation (80) is one-dimensional on the interval $y \in \big(\frac{L-1}{2L-1}, 1\big)$ (which ensures $B(y) > 0$). In practice, we solve it via bisection in that interval and then compute $\gamma_{\mathrm{WCR}}(L)$ by (81).

Representative values (computed to high precision) are:

| $L$ | 2 | 3 | 4 | 5 | 6 | 8 | 10 | 20 |
|---|---|---|---|---|---|---|---|---|
| $\gamma_{\mathrm{WCR}}(L) = \eta_{\mathrm{WCR}}/\eta_1$ | 3.196 | 3.661 | 3.979 | 4.224 | 4.427 | 4.751 | 5.008 | 5.844 |

$$\tag{82}$$

and the values continue to increase with $L$ (e.g., $\gamma_{\mathrm{WCR}}(50) \approx 7.04$, $\gamma_{\mathrm{WCR}}(100) \approx 8.02$).

### F.6. Asymptotic growth with depth

In this subsection we justify Proposition 6.3. Let $\gamma_{\mathrm{WCR}}(L) = L\alpha_{\mathrm{WCR}}(L)$, and let $y = y(L)$ be the solution of (80). Define $t := 2y - 1 \in (0, 1)$ (so $y = (1 + t)/2$). A convenient form of the boundary algebra (equivalent to (80)) is

$$B(y) = r(y)^{L-2}\big(y\,r(y)^L - 1\big), \qquad r(y) := \frac{A(y)}{B(y)}. \tag{83}$$

For large $L$, the root satisfies $t \downarrow 0$ and hence $y \downarrow 1/2$. In that regime, a standard expansion shows

$$r(y) = 1 + \frac{1}{2Lt} + o\Big(\frac{1}{Lt}\Big), \qquad \Rightarrow \qquad r(y)^L = \exp\Big(\frac{1}{2t} + o(1)\Big). \tag{84}$$

Moreover, since $B(y) = Lt + O(1)$ and $y = 1/2 + O(t)$, equation (83) implies (to leading order)

$$\exp\Big(\frac{1}{t}\Big) \asymp Lt. \tag{85}$$

Using $\gamma_{\mathrm{WCR}}(L) = L\alpha_{\mathrm{WCR}}(L)$ and $\alpha_{\mathrm{WCR}}(L) = y^{2/L-1}/B(y)$, we have $\gamma_{\mathrm{WCR}}(L) \sim 2/t$ (because $y \to 1/2$ and $B \sim Lt$). Substituting $t \sim 2/\gamma_{\mathrm{WCR}}$ into (85) yields

$$\exp\Big(\frac{\gamma_{\mathrm{WCR}}(L)}{2}\Big) \asymp \frac{L}{\gamma_{\mathrm{WCR}}(L)}, \tag{86}$$

which implies $\gamma_{\mathrm{WCR}}(L) = \Theta(\log L)$. More refined manipulations yield the Lambert-$W$ scaling

$$\gamma_{\mathrm{WCR}}(L) = 2W(\Theta(L)) = 2\log L - 2\log\log L + O(1), \tag{87}$$

establishing Proposition 6.3.

# G. Other Figures

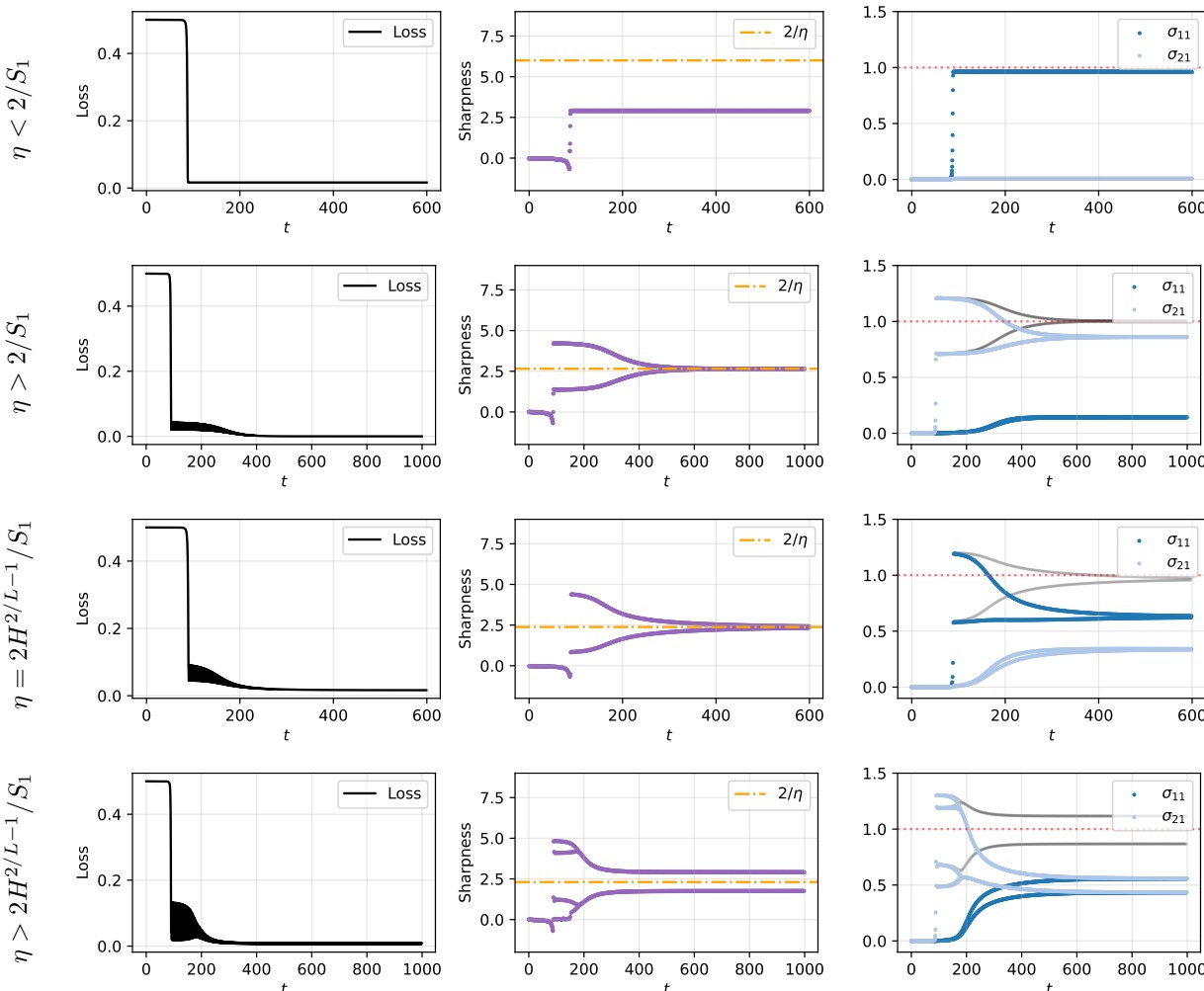

*Figure 7.* Training dynamics of GD with multi path MLP ($L = 3$, $H = 2$, $\sigma_{\star 1} = 1$) according to learning rate $\eta$. GD with larger learning rate $\eta$ mitigate sharp minima with $\lambda_{\max} > 2/\eta$.

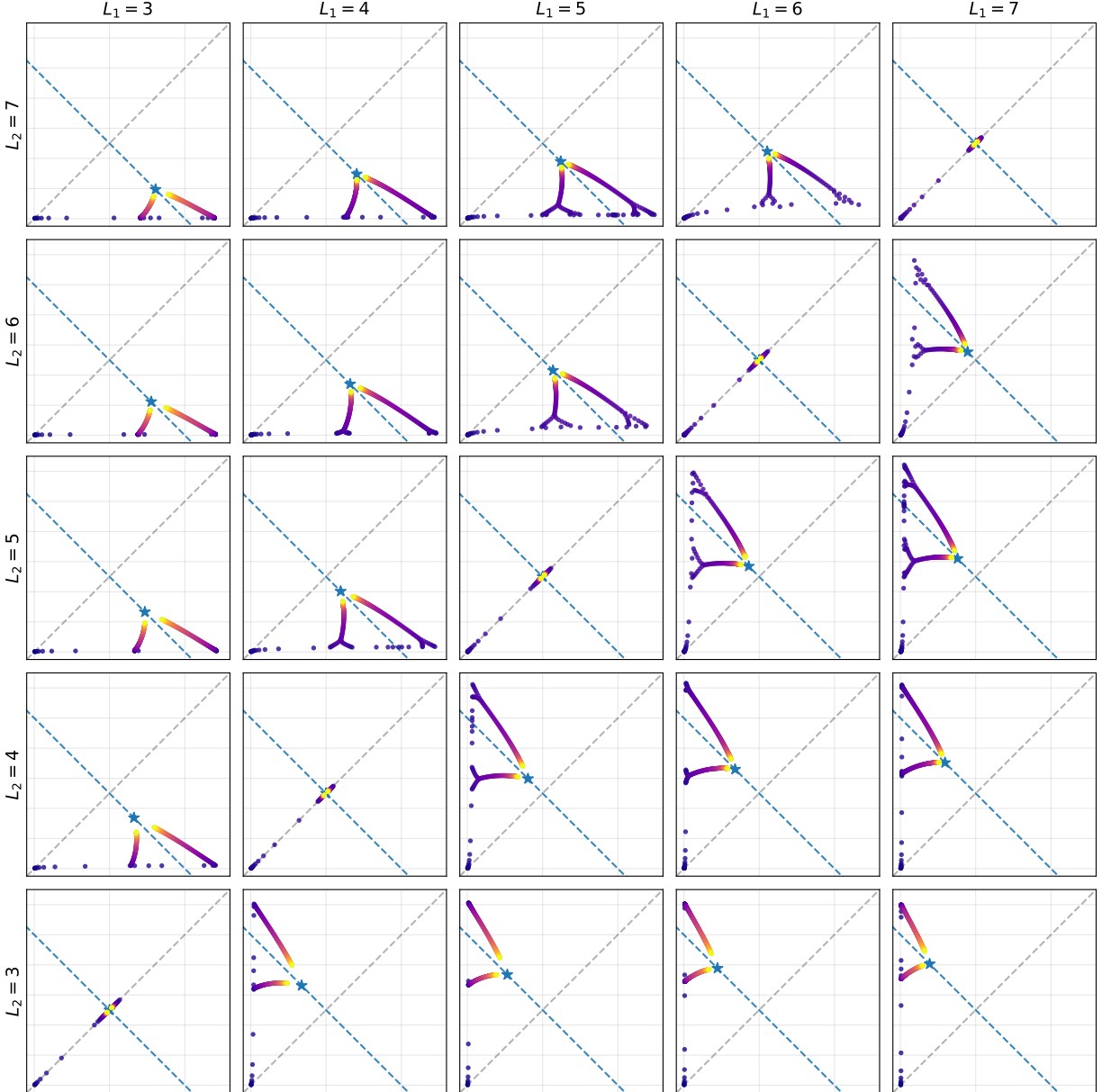

*Figure 8.* Trajectory of $(\sigma_{11}, \sigma_{21})$ for heterogeneous depth model with learning rate $\eta^*$.

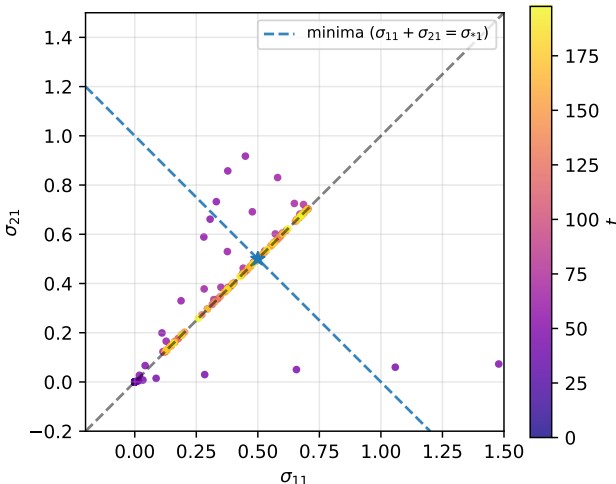

*Figure 9.* Trajectory of $\sigma_{h1}$ for learning rate $\eta = 0.99\eta_{\mathrm{WCR}}$. As learning rates getting bigger, $\sigma_{h1}$ returns close to zero.

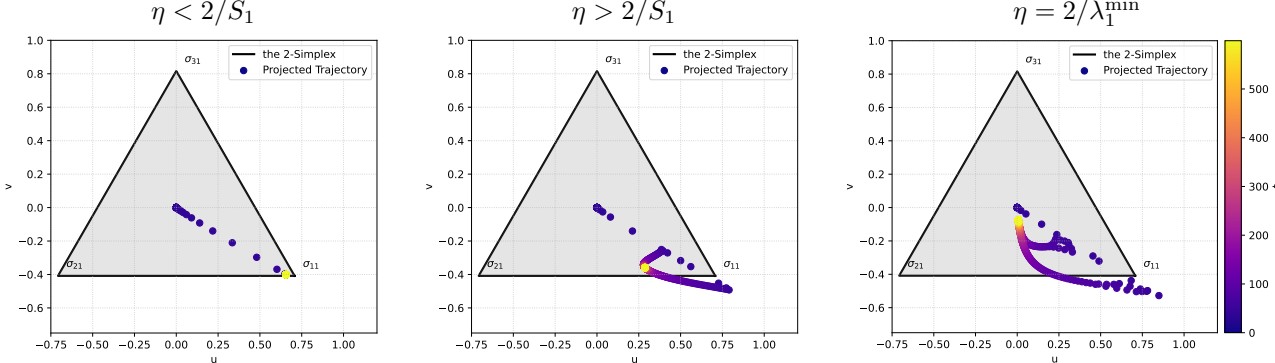

*Figure 10.* Trajecory of $\sigma_{h1}$ trained with $\eta_1 = 2/\lambda_1^{\min}$ with 2D unfolded view on minima $\sum_{h=1}^{3} \sigma_{h1} = \sigma_{\star 1}$.

