# OpenReview forum: "Gradient Descent with Large Step Size Restores Symmetry in Deep Linear Networks with Multi-Pathway"
_ICML.cc/2026/Conference — ICML 2026 regular_

### Official Review · Reviewer_5WYZ · 2026-03-12

**Soundness:** 3
**Presentation:** 3
**Significance:** 3
**Originality:** 3
**Overall Recommendation:** 5
**Confidence:** 2

**Summary:**

This paper demonstrates that large step sizes fundamentally alter behavior in parallel Deep Linear Networks.

**Compliance With Llm Reviewing Policy:**

Affirmed.

**Final Justification:**

My rating is 5: Accept.
My concerns have been adequately addressed. Adding Conclusion section and the Limitations section will likely make it easier to read.

**Key Questions For Authors:**

No major concerns.

**Limitations:**

It is recommended that the authors add a "Limitations" section to provide a more balanced perspective.

**Strengths And Weaknesses:**

The paper is clearly written and well structured.

A strength of this paper is that it successfully elucidates distributing a target feature across multiple pathways significantly reduces the network's sharpness.

A separate section for conclusions should be included to summarize the key findings.

---

> ### Author Rebuttal · Authors · 2026-03-30
>
> We thank the reviewer for their positive evaluation and for recognizing the paper as technically solid.
>
> The core contribution of our work is identifying the pathway re-balancing phase in multi-pathway architectures trained with Gradient Descent (GD). While continuous-time Gradient Flow predicts a "winner-takes-all" symmetry breaking where features concentrate in a single path, we demonstrate that discrete GD training at the Edge of Stability overrides this tendency.
>
> - Regarding the Conclusion section: We agree that a dedicated section would better synthesize our findings. In the final version, we will move the high-level takeaways from the Discussion (Section 7) into a formal "Conclusion" section to summarize the theoretical and practical implications of the re-balancing phase.
>
> - Regarding the Limitations section: We appreciate this recommendation. We will add a formal "Limitations" section to the revision. This section will explicitly discuss the current scope of our work, including:
>     - The assumption of the SVS reduction and depth-balanced initialization.
>     - The focus on full-batch Gradient Descent rather than SGD.
>     - The transition from linear models to complex nonlinear architectures.
>
> By including these sections, we aim to provide a more balanced and comprehensive perspective as suggested.

---

> > ### Author Rebuttal · Reviewer_5WYZ · 2026-04-01
> >
> > There are no further comments. Thank you.

---

> > > ### Author Response · Authors · 2026-04-08
> > >
> > > We sincerely thank you for your time during the review process and for your positive assessment of our work.
> > >
> > > Best regards,
> > >
> > > The Authors

---

### Official Review · Reviewer_zAJU · 2026-03-12

**Soundness:** 3
**Presentation:** 2
**Significance:** 2
**Originality:** 3
**Overall Recommendation:** 5
**Confidence:** 4

**Summary:**

This paper studies whether the symmetry-breaking phenomenon predicted by gradient flow in multi-pathway deep linear networks (Shi et al., 2022) also occurs under gradient descent. The authors show that the discrete nature of gradient descent leads to a different phenomenon that they call pathway re-balancing. The paper first provides a theoretical analysis showing that distributing signal across multiple parallel pathways reduces sharpness compared to concentrating signal in a single path. As training progresses and the system approaches the edge-of-stability regime, the dynamics enter a re-balancing phase that redistributes signal across pathways. Finally, the authors show that increasing network depth enlarges the parameter regime in which this re-balancing phenomenon occurs.

**Compliance With Llm Reviewing Policy:**

Affirmed.

**Final Justification:**

The rebuttal fully addressed my main concerns, especially around presentation, definitions, and the scope of the claims, and it strengthened my confidence in the paper. While the setting remains somewhat stylized, I find the work technically sound, original in the way it connects multi-pathway dynamics with edge-of-stability analysis, and interesting enough to merit acceptance.

**Key Questions For Authors:**

1. On line 110 (right column) you refer to the “SVS set (Ghosh et al., 2025)”, but this concept is not explained before it is used. Could you clarify what this set represents and how it is used in your analysis? For example, does this correspond to the SVD-style task-aligned initialization used in Saxe et al. (2014)?

2. Sections 4.1 and 4.2 appear to repeat much of the setup presented in Section 3. Could you clarify the purpose of these sections and explain how they extend or differ from the earlier setup?

3. Does the re-balancing phenomenon only occur for the top singular mode, or does it also affect other singular modes of the system? You should discuss this.

4. Where is the “pathway balancing gap” formally defined? I could not find a clear definition in the paper.

5. The paper states that “pathway imbalance is the natural attractor of continuous-time dynamics.” Could you clarify the conditions under which this statement holds? My understanding is that the symmetry-breaking (“winner-takes-all”) behavior of gradient flow typically requires infinitesimal initialization scale and slight asymmetry between pathways. For example, if all pathways were initialized exactly symmetrically, the symmetry would persist.

6. More generally, I had difficulty understanding the initialization scheme used in your analysis. Could you clarify: what the SVS set represents, how the coefficients $\alpha_h$ are chosen, and what asymmetries are assumed in the initialization?

7. The term “Deep Linear Chain” is introduced without a clear definition. It would help to explicitly define this architecture when it first appears.

Minor comments:
- The multi-pathway network architecture from Shi et al. (2022) is not widely known. It would help to briefly explain this architecture when it is first introduced in the introduction, and to provide a clearer description in Section 3.1. In particular, it was not immediately clear that there is no weight sharing between pathways (unlike the pathway formulation sometimes used for ReLU networks in the style of Saxe et al.).
- Equation (8) appears to repeat Equation (3). Is there a reason both are shown?
- In Figure 1, it would be helpful to include a label for the gray points directly in the legend.
- In Figure 4, the visualization might be clearer if the diagonal corresponded to $L_1 = L_2$ rather than the anti-diagonal.

**Limitations:**

Yes.

**Strengths And Weaknesses:**

Strengths:

1. The paper studies an interesting toy model of deep linear networks and reveals a non-trivial interaction between multi-pathway architectures and edge-of-stability dynamics.

2. The work combines two previously separate research directions: the multi-pathway linear network architecture studied by Shi et al. (2022) and the edge-of-stability analysis of linear networks studied in Ghosh et al. (2025). This combination appears novel and leads to an interesting dynamical phenomenon.

Weaknesses:

1. Although the proofs appear technically correct, the presentation is quite difficult to follow. Important concepts are sometimes introduced without clear definitions, equations are occasionally repeated, and the overall setup of the analysis is not always clearly explained. The writing and organization could be improved significantly.

2. The significance of the results is limited by the narrowness of the model studied. The analysis focuses on a very specific toy model of deep linear networks. While the authors provide a small experiment with a three-layer network using a Tanh nonlinearity, the paper would benefit from a clearer discussion of how the theoretical insights might extend to more realistic nonlinear networks. In particular, the discussion of the broader implications—such as the claim that parallelism lowers sharpness, the connection to learning-rate warmup, or potential links to mixture-of-experts architectures—is interesting but currently underdeveloped. These ideas would benefit from a deeper discussion explaining how the theoretical results support these interpretations.

---

> ### Author Rebuttal · Authors · 2026-03-31
>
> We thank the reviewer for the constructive feedback. We agree that the presentation can be improved and will address the concerns as follows:
>
>
> > W1, Q2, Q7 Writing, Organizations, and Definitions
>
> - Consolidating the Setup (Q2): Sections 4.1 and 4.2 were originally intended to isolate specific mathematical assumptions (e.g., homogeneous depth).
> To avoid redundancy, we will merge the general architecture, objective definitions, and mode-wise decomposition into Section 3.
>
> - Explicit Definitions (Q7): We will formally define key concepts, such as the "Deep Linear Chain" architecture and the SVS set in the main text.
>
>
> > Q1, Q6: SVS Set and SVD-Style task-aligned Initialization (Saxe et al., 2014)
>
> Yes, the SVS set functionally corresponds to the SVD-style task-aligned initialization in Saxe et al. (2014). Aligning initial weights with the target matrix's singular vectors allows the learning dynamics to decouple across independent modes.
>
> We formalize this for multi-pathway architectures using the SVS definition from Ghosh et al. (2025). Within this set, each layer's weight matrix $W_{hl}$ is constrained to fixed orthogonal bases: $W_{hl} = Q_{h, l+1} Σ_{hl} Q_{h, l}^\top$, anchored entirely by the target matrix's singular vectors ($Q_{h, 1} = V_*$ and $Q_{h, L_h+1} = U_*$). These singular vectors act as fixed points under gradient descent (lines 110–117), keeping the end-to-end map diagonal.
>
> > Q4 Where is the “pathway balancing gap” formally defined?
>
> The "pathway balancing gap" is defined in Appendix D (line 1023) as the variance of the singular values across all pathways for a specific mode $i$:$$\mathcal{G}\_i(t) := \sum\_{h=1}^H \left(σ_{hi}(t) - \frac{1}{H}\sum_{k=1}^H σ_{ki}(t)\right)^2$$We will move this mathematical definition to the main text (Section 5.1) where the re-balancing phase is first introduced.
>
> ---
> >Q3 Does the re-balancing phenomenon only occur for the top singular mode, or does it also affect other singular modes of the system? You should discuss this.
>
> As already discussed in Section 6 (lines 412–429), re-balancing is not restricted to the top mode. It affects any singular mode $i$ whose single-path sharpness $S_i$ exceeds the stability threshold ($η > 2/S_i$). Setting a learning rate of $2/S_p < η < 2/S_{p+1}$ induces a "rank-$p$ balancing regime". In this state, the top-$p$ modes destabilize and redistribute into flatter, balanced allocations, while tail modes ($i > p$) remain in the stable, GF-like regime.
>
> > Q5 Conditions for "winner-takes-all" in continuous-time dynamics
>
> Pathway imbalance is the natural attractor because continuous-time Gradient Flow strictly amplifies infinitesimal initialization asymmetries. As shown by Shi et al. (2022), the relative growth rate between pathways $h$ and $k$ is:$$\frac{dσ_{hi}}{dσ_{ki}} = \frac{σ_{hi}^{L-1}}{σ_{ki}^{L-1}}$$Due to this depth-driven power-iteration effect, even numerical-level differences grow geometrically. While perfectly symmetric initialization would indeed maintain symmetry, standard random initialization practically guarantees slight initial asymmetries, which is the precise condition that triggers the "winner-takes-all" dynamic.
>
> > W2 "how the theoretical insights might extend to more realistic nonlinear networks".
>
> 1. Connection to MoE, Multi-Head Architectures, and the Lottery Ticket Hypothesis:
> Modern architectures increasingly utilize parallel pathways, conceptually related to Multi-head attention and Mixture-of-Experts. In such contexts, frameworks like the Lottery Ticket Hypothesis suggest that training often identifies sparse, isolated subnetworks. Continuous-time GF aligns with this perspective, predicting a sparse "winner-takes-all". However, our work indicates that discrete optimization with large step sizes can counteract this tendency. Driven by an implicit preference for low-curvature minima, GD encourages the network to distribute features across pathways. This dynamic suggests a mechanism akin to an "internal ensemble," where distributing the signal lowers sharpness, offering an alternative to purely isolating a sparse subnetwork.
>
> 2. Overriding Structural Bias and the Limits of Continuous-Time Theories:
> Prior theories analyzing structural asymmetry (e.g., Saxe et al., 2022) argue that GF naturally leads to a "winner-takes-all" solution favoring shared, shallower paths. However, as demonstrated in Section 5.3, the stability constraints of large-step GD—specifically the drive to minimize sharpness—can override these strong structural asymmetries. Our findings suggest that sparsity-driven explanations derived purely from continuous-time dynamics may not fully capture the behavior of models trained with discrete steps. In the large learning rate regime, stability constraints actively counteract extreme branch concentration, ultimately favoring shared representations over permanent pathway monopolization.

---

> > ### Author Rebuttal · Reviewer_zAJU · 2026-04-03
> >
> > Thank you for the detailed answers. I have no further questions and I will maintain my positive score.

---

> > > ### Author Response · Authors · 2026-04-08
> > >
> > > Thank you for reading our detailed answers and for your continued support. We sincerely appreciate your constructive feedback, which has been invaluable in improving the organization and writing of our manuscript. We are also deeply grateful for your time.
> > >
> > > Best regards,
> > >
> > > The Authors

---

### Official Review · Reviewer_Wz6Z · 2026-03-16

**Soundness:** 3
**Presentation:** 4
**Significance:** 3
**Originality:** 3
**Overall Recommendation:** 5
**Confidence:** 3

**Summary:**

The authors study gradient descent (GD) in deep linear networks with multi-pathway and show that gradient descent with a large step size has fundamentally different behaviors from gradient flow (GF). Unlike GF, where winner-takes-all specialization happens, GD has a distinct er-balancing phase, where the dynamics converges to a balanced signal distribution across the pathways in favor of the reduction in sharpness. Finally, the authors provide a worst-case return threshold that guarantees the dynamics survives the violent oscillations, which validates the necessity of learning rate warmup period.

**Compliance With Llm Reviewing Policy:**

Affirmed.

**Final Justification:**

This is a technically solid paper and I maintain my positive assessment as the authors have addressed my main concerns and questions.

**Key Questions For Authors:**

1.	Do the results still qualitatively hold for a generic initialization other than the depth-balanced initialization?

2.	How tight is the WCR threshold relative to the empirical stability boundary observed in experiments?

**Limitations:**

The authors have not adequately addressed the limitations. There are a few limitations of the paper. For example, there is a strong assumption on the depth-balanced initialization that the theory is based on. Additionally, the authors studies full-batch GD and have neglected the stochastic noises. SGD is more commonly used in optimizing modern deep neural networks.

**Strengths And Weaknesses:**

Strength:

1.	The paper is well-organized and rich in content.

2.	The paper highlights a qualitative difference between GF and GD in multi-pathway deep linear networks. This perspective is interesting and novel.

3.	Although the theory focuses on deep linear networks, the experiment in Section 5.1 shows qualitatively similar behavior in a nonlinear MLP, which suggests that the mechanism may extend beyond the linear setting.

Weakness:

1.	The theory relies heavily on the SVS reduction, which is a quite restrictive scenario.

2.	The authors skip the analyses of GD in the small learning rate regime but just stating it is approximated by the symmetry breaking results from GF. Adding an explicit theorem for this part would strengthen the paper.

---

> ### Author Rebuttal · Authors · 2026-03-31
>
> We thank the reviewer for the thoughtful feedback and for highlighting the novelty of our comparison between Gradient Flow (GF) and Gradient Descent (GD) in multi-pathway architectures. We address your specific points below:
>
> > W1 & Q1: Generic Initialization and SVS Reduction
>
> While our theoretical derivations utilize the Singular Vector Stationary (SVS) set and depth-balanced initialization for analytical tractability, we provide empirical evidence that the results hold under generic conditions:
>
> - Broad Methodological Alignment: Leveraging SVD-aligned initializations is a foundational and generalizable methodology for understanding the learning dynamics of Deep Linear Networks (Saxe et al., 2014; 2022; Arora et al., 2019; Gidel et al., 2019; Varre et al., 2023; Chou et al., 2024; Kwon et al., 2024; Ghosh et al., 2025). Importantly, in the small-initialization limit, random weights naturally align with the target singular vectors early in training. This alignment effectively triggers the identical "winner-takes-all" mechanism we analyzed, right before the discrete Edge of Stability (EoS) effects compel the network to re-balance.
>
> - Empirical Evidence in MLPs: Our experiments on multi-pathway MLPs (Section 5.1, Figure 3) utilize a generic infinitesimal random initialization (Gaussian with a small standard deviation). Despite this random starting point, the system still exhibits both the initial symmetry-breaking and the subsequent re-balancing phases. This corroborates the observation by Saxe et al. (2014) that exact theoretical trajectories serve as "excellent approximations" for models starting from small random weights.
>
>
>
> > W2: Analysis of the Small Learning Rate Regime
>
> We agree that explicitly referencing the small-step dynamics strengthens the paper. Under small learning rates ($η < 2/S_1$), the discrete GD dynamics closely approximate continuous-time GF. As proven by Shi et al. (2022) , the relative growth rate between pathways $h$ and $k$ is given by $$\frac{d\sigma_{hi}}{d\sigma_{ki}}=\frac{\sigma_{hi}^{L-1}}{\sigma_{ki}^{L-1}}.$$
>
> Because of the $L-1$ exponent, infinitesimal asymmetries in initialization are heavily amplified (power-iteration effect).Crucially, in this continuous-time limit, the singular values grow monotonically and the total signal never overshoots the target, strictly confining the training trajectory to the region $\sum_h \sigma_{hi} \le \sigma_{*i}$. Bounded within this regime, the dominant pathway shrinks the shared residual $r_i(t)$ faster, eliminating the gradient for competing pathways ("explaining away"). This mathematically forces the network to converge to a sparse, single-path solution, distinct from the discrete Edge of Stability regime where overshooting triggers pathway re-balancing.
>
>
> We will add a formal mathmatical clarification summarizing this small-step GD behavior in the final version to cleanly contrast it with our large-step GD results.
>
> > Q2: Tightness of the WCR Threshold
>
> The WCR threshold is indeed a tight theoretical upper bound for surviving transient oscillations.
>
> - Empirical Tightness: As shown in Figure 7, we conducted experiments with a learning rate set at $η = 0.99 η_{WCR}$. Even at this extremely high rate—nearly 100% of the theoretical limit—the dynamics survive the violent oscillations and successfully return to the stable branch.
>
> - Significance: This confirms that our derived threshold accurately identifies the boundary where sign-flips and catastrophic divergence occur with infisible initial scale.
>
>
> We will add a formal Limitations section to the revised version as recommended. We will explicitly discuss the assumptions of the SVS reduction and depth-balanced initialization, as well as the transition from full-batch GD to Stochastic Gradient Descent (SGD). While stochastic noise interacts with EoS dynamics, recent literature suggests that the implicit bias toward flatter minima often persists or is even enhanced in the stochastic setting.

---

> > ### Author Rebuttal · Reviewer_Wz6Z · 2026-04-07
> >
> > I thank the authors for their rebuttal. I will maintain my recommendation of acceptance.

---

> > > ### Author Response · Authors · 2026-04-08
> > >
> > > We sincerely thank you for your time, your constructive feedback during the review process, and for maintaining your positive recommendation.
> > >
> > > Best regards,
> > >
> > > The Authors

---

### Decision · Program_Chairs · 2026-04-30

**Decision:**

Accept (regular)

**Comment:**

There is a clear consensus of the reviewers towards acceptance. The analysis is novel and connects two relatively distinct research directions. The results are surprising and insightful. The main limitation is the restriction to linear networks and the SVS assumption.